# RILe: Reinforced Imitation Learning

## Abstract

Acquiring complex behaviors is essential for artificially intelligent agents, yet learning these behaviors in high-dimensional settings poses a significant challenge due to the vast search space. Traditional reinforcement learning (RL) requires extensive manual effort for reward function engineering. Inverse reinforcement learning (IRL) uncovers reward functions from expert demonstrations but relies on an iterative process that is often computationally expensive. Imitation learning (IL) provides a more efficient alternative by directly comparing an agent's actions to expert demonstrations; however, in high-dimensional environments, such direct comparisons often offer insufficient feedback for effective learning. We introduce RILe (Reinforced Imitation Learning), a framework that combines the strengths of imitation learning and inverse reinforcement learning to learn a dense reward function efficiently and achieve strong performance in high-dimensional tasks. RILe employs a novel trainer–student framework: the trainer learns an adaptive reward function, and the student uses this reward signal to imitate expert behaviors. By dynamically adjusting its guidance as the student evolves, the trainer provides nuanced feedback across different phases of learning. Our framework produces high-performing policies in high-dimensional tasks where direct imitation fails to replicate complex behaviors. We validate RILe in challenging robotic locomotion tasks, demonstrating that it significantly outperforms existing methods and achieves near-expert performance across multiple settings.

## 1 Introduction

Learning complex behaviors is critical for advancing artificially intelligent agents in fields such as robotics and strategic games. Over the years, reinforcement learning (RL) has emerged as a powerful framework for teaching agents to perform sophisticated tasks, yet it often requires extensive manual reward function design. This is both time-consuming and error-prone.

There are two ways to address the reward engineering problem. First, Inverse Reinforcement Learning (IRL) (Ng & Russell, 2000; Ziebart et al., 2008) offers a remedy by inferring the reward function from expert demonstrations, thus reducing the burden of manual reward engineering. IRL proceeds iteratively: it first trains a policy (the learning agent's decision-making mechanism) using the current reward function, observes how well the agent's behavior aligns with the expert's, and then refines the reward function to better guide the policy toward expert-like behaviors. Repeating this process eventually yields a reward function capable of providing nuanced feedback at different stages of learning. However, this iterative procedure is computationally expensive (Zheng et al., 2022), especially in high-dimensional environments where both the reward and the policy must explore a large state-action space.

Second, Imitation learning (IL) bypasses explicit reward design by directly comparing learned behaviors to expert demonstrations via a comparison mechanism. Traditional IL approaches such as Behavioral Cloning (BC) (Bain & Sammut, 1995) match the learned actions to expert demonstrations directly, requiring a substantial amount of expert data in high-dimensional tasks. Adversarial Imitation Learning (AIL) methods, such as GAIL (Ho & Ermon, 2016), enables policy learning via RL by introducing a discriminator as a comparison mechanism that judges how expert-like the learned behaviors are. However, both traditional IL and AIL lack a reward function that emphasizes specific subgoals or partial improvements, or boosts exploration. Instead, they rely on a distance measure or a binary classifier, that merely checks whether the agent's behavior is (or is not) similar to the expert. Such comparison-based signals offer no fine-grained

guidance on which specific actions or sub-strategies to prioritize, or which actions to explore. Consequently, both traditional IL and AIL struggle in high-dimensional environments (Peng et al., 2018; Garg et al., 2021), where the agent needs more granular and adaptive feedback than these mechanisms provide.

Adversarial Inverse Reinforcement Learning (AIRL) (Fu et al., 2018) attempts to remedy IRL's inefficiency by integrating a learned reward function within a discriminator. However, AIRL tightly couples the reward function to the discriminator's output, causing it to inherit AIL's limitations in high-dimensional settings where more fine-grained guidance is needed.

In contrast, real-life learning scenarios suggest a different approach: think of parents and children, or a pet owner and their dog. The *teacher* also refines how they teach as the student progresses. Each success or failure in the student's understanding informs the teacher's approach, creating a *positive-sum* relationship: lessons learned from suboptimal behaviors ultimately yield better trainers, which, in turn, guide the student more effectively. By contrast, existing approaches lack this cooperative synergy. Adversarial Imitation Learning (AIL) does update a discriminator alongside the policy, but the discriminator's sole role is to distinguish expert-like behavior from non-expert behavior. Consequently, the student attempts to fool this judge into classifying its behavior as expert-like, resulting in a *competitive* process rather than a *cooperative trainer* that dynamically shapes rewards based on suboptimal behaviors. Meanwhile, IRL methods only refine the reward *after* the policy converges, missing the opportunity for continuous co-evolution throughout training.

To address these issues, inspired by these insights, we propose Reinforced Imitation Learning (RILe). RILe combines the adaptive reward benefits of IRL with the computational efficiency of AIL (Fig. 1-(d)). RILe is a novel *trainer-student* system that establishes a positive-sum relationship between the trainer and the student. Specifically, RILe is composed of:

- **Student Agent:** Learns a policy to imitate expert demonstrations using reinforcement learning.
- **Trainer Agent:** Simultaneously learns a reward function using reinforcement learning, leveraging an adversarial discriminator for continuous feedback on student performance.

RILe's trainer continuously updates the reward function in tandem with the student's policy updates, where IRL refines its reward function only after training a policy to convergence on the current reward function. Specifically, the trainer queries a discriminator to measure how expert-like the student's behavior is, then optimizes the reward function based on that feedback, without waiting for the policy to converge. RILe offers nuanced reward shaping, while avoiding IRL's heavy computational loop. As a result, RILe is particularly effective in high-dimensional settings, where agents need fine-grained guidance at every stage of learning. Our contributions are two-fold:

1. **Efficient Reward-Function Learning via RL**: We introduce a reinforcement-learning-based approach for training a reward function simultaneously with the policy. This avoids IRL's repeated policy re-training and the purely discriminator-based rewards of AIL/AIRL. Unlike the competitive judge in AIL, using RL allows RILe's trainer agent to explore reward strategies and learn adaptively from the student's progress, establishing a cooperative dynamic that yields a reward function that is optimized for a long-horizon objective.
2. **Dynamic Reward Customization:** RILe offers context-sensitive guidance at every stage of training, because the trainer agent updates the reward function as the student evolves. This dynamic shaping is especially valuable in high-dimensional tasks, where the learning agent requires different forms of encouragement during intermediate-stages than later-stages of the training. Consequently, RILe enables accurate imitation of the expert performance in high-dimensional tasks.

We evaluate RILe in comparison to state-of-the-art methods in AIL, IRL, and AIRL: GAIL (Ho & Ermon, 2016) AIRL (Fu et al., 2018), GAIfO (Torabi et al., 2018b), BCO (Torabi et al., 2018a), IQ-Learn (Garg et al., 2021) and DRAIL (Lai et al., 2024). Our experiments span six studies: (1) Empirically analyzing how RILe's reward-learning differs from baselines, (2) Quantitatively analyzing the learned reward function in RILe, (3) Comparing different trainer-discriminator relationships in RILe, (4) Evaluating the noise robustness of RILe, (5) Analyzing the impact of using expert-data explicitly inside RILe, and (6) Assessing RILe's performance in both low- and high-dimensional continuous-control problems. Our results show RILe's

superior performance, particularly in high-dimensional environments, and highlight RILe's ability to learn a dynamic reward function that effectively guides the student through multiple stages of training.

## 2 Related Work

We review research on learning from expert demonstrations, focusing on Imitation Learning (IL) and Inverse Reinforcement Learning (IRL), the conceptual foundations of RILe.

**Imitation Learning** Early work in IL introduced Behavioral Cloning (BC) (Bain & Sammut, 1995), which frames policy learning as a supervised problem where the agent's actions are directly matched to expert demonstrations. DAgger (Ross et al., 2011) refines BC by aggregating data over multiple iterations to mitigate compounding errors. GAIL (Ho & Ermon, 2016) employs adversarial training: a discriminator learns to distinguish expert trajectories from the agent's, while the generator (agent) adapts to mimic expert-like behavior. BCO (Torabi et al., 2018a) extends BC, and GAIfO (Torabi et al., 2018b) extends GAIL, both to handle state-only observation scenarios. DQfD (Hester et al., 2018) introduces a two-stage approach with pre-training, while ValueDice (Kostrikov et al., 2020) aligns policy and expert distributions via a distribution-matching objective. More recently, DRAIL (Lai et al., 2024) leverages a diffusion-based discriminator to enhance learning efficiency in adversarial imitation.

Despite these advances, IL methods face challenges in high-dimensional environments (Peng et al., 2018; Garg et al., 2021), where naive action matching or purely adversarial comparisons fail to provide sufficiently granular guidance. RILe addresses these limitations through an adaptive trainer–student framework, where a learned reward function provides more nuanced guidance than standard IL comparison mechanisms.

**Inverse Reinforcement Learning** Inverse Reinforcement Learning (IRL), introduced by Ng & Russell (2000), aims to uncover the expert's intrinsic reward function from demonstrations. Major developments include Apprenticeship Learning (Abbeel & Ng, 2004), Maximum Entropy IRL (Ziebart et al., 2008), and adversarial variants like AIRL (Fu et al., 2018). IQ-Learn (Garg et al., 2021) reformulates IRL by integrating the inverse reward learning process into Q-learning for better scalability. More recent work focuses on unstructured data (Chen et al., 2021) and cross-embodiment transfer (Zakka et al., 2022).

Nonetheless, IRL methods struggle with computational inefficiency and limited scalability (Arora & Doshi, 2021), particularly in high-dimensional tasks where repeated iterations of policy learning and reward refinement become costly. RILe mitigates these challenges by jointly learning the policy and reward function in a single process, avoiding IRL's iterative retraining loop and facilitating more efficient reward shaping for complex environments.

## 3 Background

### 3.1 Markov Decision Process

A standard Markov Decision Process (MDP) is defined by $(S, A, R, T, K, \gamma)$. $S$ is the state space consisting of all possible environment states $s$, and $A$ is action space containing all possible environment actions $a$. $R = R(s, a) : S \times A \rightarrow \mathbb{R}$ is the reward function. $T = \{P(\cdot|s, a)\}$ is the transition dynamics where $P(\cdot|s, a)$ is an unknown state state transition probability function upon taking action $a \in A$ in state $s \in S$. $K(s)$ is the initial state distribution, i.e., $s_0 \sim K(s)$ and $\gamma$ is the discount factor. The policy $\pi = \pi(a|s) : S \rightarrow A$ is a mapping from states to actions. In this work, we consider $\gamma$-discounted infinite horizon settings. Following Ho & Ermon (2016), expectation with respect to the policy $\pi \in \Pi$ refers to the expectation when actions are sampled from $\pi(s)$: $\mathbb{E}_\pi[R(s, a)] \triangleq \mathbb{E}_\pi[\sum_{t=0}^{\infty} \gamma^t R(s_t, a_t)]$, where $s_0$ is sampled from an initial state distribution K(s), $a_t$ is given by $\pi(\cdot|s_t)$ and $s_{t+1}$ is determined by the unknown transition model as $P(\cdot|s_t, a_t)$. The unknown reward function $R(s, a)$ generates a reward given a state-action pair $(s, a)$. We consider a setting where $R = R(s, a)$ is parameterized by $\theta$ as $R_\theta(s, a) \in \mathbb{R}$ (Finn et al., 2016).

Our work considers an imitation learning problem from expert trajectories, consisting of states $s$ and actions $a$. The set of expert trajectories $\tau_E$ are sampled from an expert policy $\pi_E \in \Pi$, where $\Pi$ is the set of all

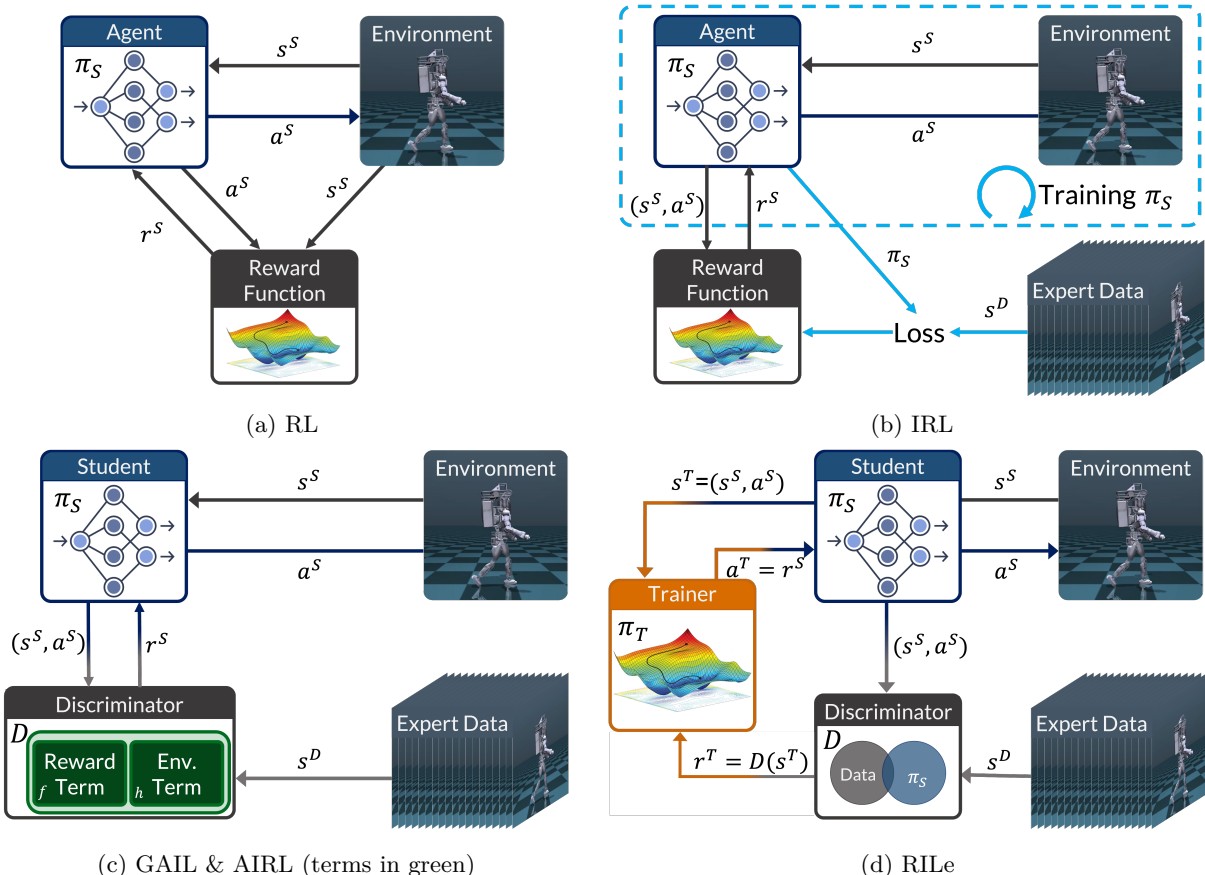

(a) RL

(b) IRL

(c) GAIL & AIRL (terms in green)

(d) RILe

Figure 1: **Overview of the related works. (a) Reinforcement Learning (RL):** learning a policy that maximizes hand-defined reward function; **(b) Inverse RL (IRL):** learning a reward function from data. IRL has two stages: 1. training a policy with frozen reward function, and 2. updating the reward function by comparing the converged policy with data. These stages repeated several times; **(C) Generative Adversarial Imitation Learning (GAIL) & Adversarial IRL (AIRL):** using discriminator as a reward function. GAIL trains both policy and the discriminator at the same time. AIRL implements a new structure on the discriminator, seperating reward from environment dynamics by using two networks under the discriminator (see additional terms in green). **(D) RILe:** similar to IRL, learning a reward function from data. RILe learns the reward function at the same time with the policy, using a discriminator as a guide for learning the reward.

possible policies. We assume that we have access to $m$ expert trajectories, all of which have $n$ time-steps, $\tau_E = \{(s_0^i, a_0^i), (s_1^i, a_1^i), \dots, (s_n^i, a_n^i)\}_{i=1}^m$.

### 3.2 Reinforcement Learning (RL)

Reinforcement learning seeks an optimal policy, $\pi^*$. that maximizes the discounted cumulative reward from the reward function $R = R(s, a)$ (Fig. 1-(a)). In this work, we incorporate entropy regularization using the $\gamma$-discounted casual entropy function $H(\pi) = \mathbb{E}_\pi[-\log \pi(a|s)]$ (Ho & Ermon, 2016; Bloem & Bambos, 2014). The RL problem with a parameterized reward function and entropy regularization is defined as

$$\text{RL}(R_\theta(s, a)) = \pi^* = \arg\max_\pi \mathbb{E}_\pi[R_\theta(s, a)] + H(\pi). \tag{1}$$

### 3.3 Inverse Reinforcement Learning (IRL)

Given sample trajectories $\tau_E$ from an optimal expert policy $\pi_E$, inverse reinforcement learning aims to recover a reward function $R_\theta^*(s, a)$ that maximally rewards the expert's behavior (Fig. 1-(b)). Formally, IRL seeks a reward function, $R_\theta^*(s, a)$, satisfying: $\mathbb{E}_{\pi_E}[\sum_{t=0}^\infty \gamma^t R_\theta^*(s_t, a_t)] \geq \mathbb{E}_\pi[\sum_{t=0}^\infty \gamma^t R_\theta^*(s_t, a_t) + H(\pi)] \quad \forall \pi$. Optimizing this reward function with reinforcement learning yields a policy that replicates expert behavior: $\mathrm{RL}(R_\theta^*(s, a)) = \pi^*$. Since only the expert's trajectories are observed, expectations over $\pi_E$ are estimated from samples in $\tau_E$. Incorporating entropy regularization $H(\pi)$, maximum causal entropy inverse reinforcement learning (Ziebart et al., 2008) is defined as

$$\mathrm{IRL}(\tau_E) = \underset{R_\theta(s, a) \in \mathbb{R}}{\arg\max} \left( \mathbb{E}_{s, a \in \tau_E}[R_\theta(s, a)] - \max_\pi \left( \mathbb{E}_\pi[R_\theta(s, a)] + H(\pi) \right) \right). \tag{2}$$

### 3.4 Adversarial Imitation Learning (AIL) and Adversarial Inverse Reinforcement Learning (AIRL)

Imitation Learning (IL) aims to directly approximate the expert policy from given expert trajectory samples $\tau_E$. It can be formulated as $\mathrm{IL}(\tau_E) = \arg\min_\pi \mathbb{E}_{(s,a)\sim\tau_E}[L(\pi(\cdot|s), a)]$, where $L$ is a loss function, that captures the difference between policy and expert data.

GAIL (Ho & Ermon, 2016) introduces an adversarial imitation learning setting by quantifying the difference between the agent and the expert with a discriminator $D_\phi(s, a)$, parameterized by $\phi$ (Fig. 1-(c)). The discriminator distinguishes between between expert-generated state-action pairs $(s, a)$ sampled from the expert trajectories $\tau_E$ and non-expert ones $(s, a) \notin \tau_E$. The goal of GAIL is to find the optimal policy that fools the discriminator while maximizing an entropy constraint. The optimization is formulated as a zero-sum game between the discriminator $D_\phi(s, a)$ and the policy $\pi$:

$$\min_\pi \max_\phi \mathbb{E}_\pi[\log D_\phi(s, a)] + \mathbb{E}_{\tau_E}[\log (1 - D_\phi(s, a))] - \lambda H(\pi). \tag{3}$$

In other words, the reward function that is maximized by the policy is defined as a similarity function, expressed as $R(s, a) = -\log (D_\phi(s, a))$.

AIRL (Fu et al., 2018) extends AIL to inverse reinforcement learning, aiming to recover a reward function decoupled from environment dynamics (Fig. 1-(c)). AIRL structures the discriminator as:

$$D_{\phi,\psi}(s, a, s') = \frac{\exp(f_\phi(s, a, s'))}{\exp(f_\phi(s, a, s')) + \pi(a|s)}, \tag{4}$$

where $f_\phi(s, a, s') = r_\psi(s, a) + \gamma V_\phi(s') - V_\phi(s)$. Here, $r_\psi(s, a)$ represents the learned reward function that is decoupled from the environment dynamics, and $\gamma V_\phi(s') - V_\phi(s)$ is a discriminator based shaping term. The AIRL optimization problem is formulated equivalently to GAIL (see Eqn. 3). The reward function $r_\psi(s, a)$ is learned through minimizing the cross-entropy loss inherent in this adversarial setup. Therefore, the reward function remains tightly coupled with the discriminator's learning process.

## 4 RILe: Reinforced Imitation Learning

We propose Reinforced Imitation Learning (RILe) to jointly learn a reward function and a policy that emulates expert-like behavior within a single learning process. RILe introduces a novel trainer–student dynamic, as illustrated in Figure 2.

In RILe, the student agent learns an action policy by interacting with the environment, while the trainer agent learns a reward function that effectively guides the student toward expert-like behavior. Both agents are trained simultaneously via reinforcement learning, with an assistance from an adversarial discriminator. Specifically, the trainer queries the discriminator, which judges how expert-like the student's behavior is, and then optimizes the reward function based on that feedback on-the-fly. Unlike traditional AIL, where the discriminator effectively is employed as the reward function for the student, RILe introduces a trainer agent to provide fine-grained feedback to the student, while avoiding IRL's iterative computational expense.

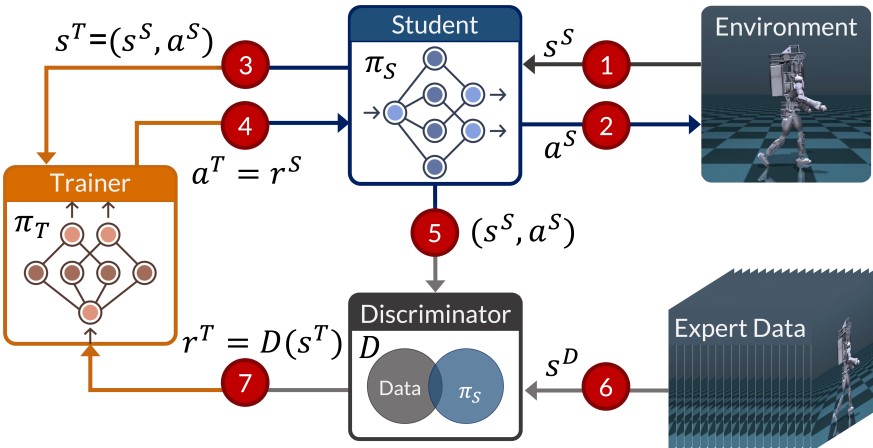

Figure 2: **Reinforced Imitation Learning (RILe)**. The framework consists of three key components: a student agent, a trainer agent, and a discriminator. The student agent learns a policy $\pi_S$ by interacting with an environment, and the trainer agent learns a reward function as a policy $\pi_T$. (1) The student receives the environment state $s^S$. (2) The student takes an action $a^S$, forwards it to the environment which is updated based on $a^S$. (3) The student forwards its state and action to the trainer, whose state is $s^T = (s^S, a^S)$. (4) Trainer, $\pi_T$, evaluates the state action pair of the student agent $s^T = (s^S, a^S)$ and chooses an action $a^T$ that then becomes the reward of the student agent $a^T = r^S$. (5) Discriminator takes the state-action pair of the student the $(s^S, a^S)$ (6) Discriminator compares student state-action pair with expert demonstrations $(s^D)$. (7) Discriminator gives reward to the trainer, based on the similarity between student- and expert-behavior.

The trainer agent plays the key role in RILe. Trained via RL, the trainer explores different reward designs and learns to provide gradually tailored feedback to the student by maximizing the cumulative rewards it receives from the discriminator. This approach equips RILe with two key advantages over existing IRL/AIRL/AIL frameworks: (1) On-the-fly reward function learning via RL: The reward function is learned continuously with RL, enabling the trainer to explore different reward options and account for long-horizon effects of its signals, (2) Context-sensitive guidance: The trainer adjusts its reward outputs in response to the student's current policy, thereby encouraging the student to learn actions that ultimately guide it closer to expert behavior. By providing tailored feedback at different stages of training, RILe addresses the limitations of prior methods, particularly in high-dimensional tasks (see Appendix B for more discussion).

In the remainder of this section, we define the components of RILe and explain how they jointly learn from expert demonstrations.

**Student Agent**   The student agent learns a policy $\pi_S$ by interacting with an environment in a standard RL setting within an MDP. For each of its actions $a^S \in A$, the environment returns a new state $s^S \in S$. However, instead of using a handcrafted reward function, the student's reward comes from the trainer agent's policy, $\pi_T$. Therefore, the reward function is represented by the trainer policy. Thus, the student agent is guided by the actions of the trainer agent, i.e., the action of the trainer is the reward of the student: $r^S = \pi_T((s^S, a^S))$. The optimization problem of the student agent is then defined as

$$\max_{\pi_S} \mathbb{E}_{(s^S, a^S) \sim \pi_S}[\pi_T((s^S, a^S))] + \alpha H(\pi_S). \tag{5}$$

**Discriminator**   The discriminator differentiates between expert-generated state-action pairs, $(s, a) \sim \tau_E$, and pairs from the student, $(s, a) \sim \pi_S$. In RILe, the discriminator is defined as a feed-forward deep neural network, parameterized by $\phi$. Its objective is:

$$\max_{\phi} \mathbb{E}_{(s,a) \sim \tau_E}[\log(D_\phi(s,a))] + \mathbb{E}_{(s,a) \sim \pi_S}[\log(1 - D_\phi(s,a))]. \tag{6}$$

To provide effective guidance, the discriminator must accurately identify whether a given state–action pair originates from the expert distribution $(s, a) \sim \tau_E$ or not $(s, a) \notin \tau_E$. GAIL (Ho & Ermon, 2016) established the feasibility of such a discriminator (see Appendix B for details).

**Trainer Agent** The trainer agent guides the student toward expert behavior by serving as its reward mechanism. Since the trainer does not directly observe the student's policy $\pi_S$, we model the trainer's environment as a Partially Observable MDP (POMDP): $\text{POMDP}_T = (S_T, A_T, \Omega_T, T_T, O_T, R_T, \gamma)$. The state space $S_T = S \times A \times \pi_S$ includes all possible state-action pairs from the standard MDP and the student's policy $\pi_S$, which is hidden from the trainer, introducing partial observability. The trainer's action space, $A_T$, consists of scalar values. Formally, $A_T$ is $\mathbb{R}$. The observation space $\Omega_T = S \times A$ consists of the observable state-action pairs of the student. The transition dynamics $T_T$ and the observation function $O_T$ are defined formally in Appendix A. The reward function $R_T(s^T, a^T)$ evaluates the effectiveness of the trainer's action in guiding the student, where $s^T = (s^S, a^S)$ is the observation of the trainer. $\gamma$ is the discount factor.

Within this POMDP, the trainer learns a policy $\pi_T$ that produces helpful reward signals for $\pi_S$. The trainer observes only the student's state–action pair, $s^T = (s^S, a^S) \in S \times A$, not $\pi_S$ itself. It then outputs a scalar action $a^T \in [-1, 1]$, which is provided to the student as the reward, $r^S$.

If the trainer's reward depends only on the discriminator's output, the trainer receives the same reward regardless of whether it rewards or penalizes the student, yielding no immediate feedback on its choices. For instance, if the student behaves like the expert and the discriminator outputs $\approx 1$, the trainer should ideally reward the student (action, $a^T, \approx 1$). But if the trainer's action is not factored into its own reward, it gains no immediate signal whether rewarding or punishing the student was effective, since it receives the same reward in either case. This ambiguity forces extensive trial and error. To address this, we define the trainer reward as:

$$R^T = e^{-|v(D_\phi(s^T)) - a^T|} \tag{7}$$

where $v(x) = 2x - 1$ scales the discriminator's output, making it symmetric around zero. Including $a^T$ in the trainer's reward ensures the trainer effectively learns from its own actions. Formally, we define the trainer's objective as:

$$\max_{\pi_T} \mathbb{E}_{\substack{(s,a) \sim \pi_S \\ a^T \sim \pi_T}} [e^{-|v(D_\phi(s^T)) - a^T|}] + \alpha H(\pi_T). \tag{8}$$

**RILe** RILe brings together these three components, student, trainer, and discriminator, to discover a student policy that imitates expert behaviors in $\tau_E$. Both $\pi_S$ and $\pi_T$ can be trained via any single-agent RL method. The overall training algorithm is detailed in Appendix K.

The student agent aims to recover the optimal policy $\pi_S^*$:

$$\pi_S^* = \arg\max_{\pi_S} \mathbb{E}_{(s^S, a^S) \sim \pi_S} \left[ \sum_{t=0}^{\infty} \gamma^t [\pi_T ((s_t^S, a_t^S)) + \alpha H(\pi_S(\cdot|s_t^S))] \right]. \tag{9}$$

Simultaneously, the trainer aims to recover $\pi_T^*$:

$$\pi_T^* = \arg\max_{\pi_T} \mathbb{E}_{\substack{s^T \sim \pi_S \\ a^T \sim \pi_T}} \left[ \sum_{t=0}^{\infty} \gamma^t [e^{-|v(D_\phi(s_t^T)) - a_t^T|} + \alpha H(\pi_T(\cdot|s_t^T))] \right]. \tag{10}$$

By optimizing these objectives together, RILe efficiently learns both a reward function and a policy in high-dimensional settings where traditional AIL or IRL methods often struggle. Achieving stable joint training requires specific techniques in RILe's adaptive system, which are discussed in Appendix C.

## 5 Experiments

We evaluate the performance of RILe in this section. We perform five ablation studies to analyze RILe:

1. **Reward Function Evaluation:** We qualitatively analyze how RILe's reward-learning strategy differs from baselines.
2. **Reward Function Dynamics:** We quantitatively assess the reward function learned by RILe.
3. **Trainer-Discriminator Relation:** We compare different trainer reward functions and investigate how the interaction between the trainer and the discriminator affects RILe's performance

(a) RILe

(b) GAIL

(c) AIRL

Figure 3: **Reward Function Comparison**. Evolution of reward functions during training for (a) RILe, (b) GAIL, and (c) AIRL in a continuous maze environment. Columns show reward landscapes at 25%, 50%, 75%, and 100% of training completion (left to right). The expert's trajectory is shown in red, while the student agent's trajectory from the previous training epoch is in black. Color gradients represent reward values, with darker colors indicating lower rewards and brighter colors indicating higher rewards. Black squares represent obstacles. RILe demonstrates a dynamic reward function that adapts with the student's progress, while GAIL and AIRL maintain relatively static reward landscapes throughout training and struggle to adapt.

4. **Robustness to Noise and Covariance Shift:** We investigate the robustness of RILe to different types of noise and covariate shift in the environment.

5. **Explicit Usage of Expert Data**: Analyzing the effect of using expert data explicitly inside RILe on the trainer-student dynamics.

Then, we evaluate the performance of RILe in high-dimensional tasks with two experiments:

1. **Learning from Expert Demonstrations**: Learning from perfect expert demonstrations in a reinforcement learning benchmark.

2. **Robotic Continuous Control from Motion Capture Data**: Learning to walk and to carry a box from motion-capture data with four different embodiments with unique configurations.

**Baselines** We compare RILe with seven baseline methods: Behavioral cloning (BC (Bain & Sammut, 1995; Ross & Bagnell, 2010), BCO (Torabi et al., 2018a)), adversarial imitation learning (GAIL (Ho & Ermon, 2016), GAIfO (Torabi et al., 2018b) and DRAIL (Lai et al., 2024)), adversarial inverse reinforcement learning (AIRL (Fu et al., 2018)), and inverse reinforcement learning (IQ-Learn (Garg et al., 2021)). DRAIL (Lai et al., 2024) introduces a diffusion-based discriminator implementation, which is applied to both GAIL and RILe, and referred as DRAIL-GAIL and DRAIL-RILe.

Additional experimental details are provided in the Appendix D, and hyperparameter selections are discussed in the Appendix G.

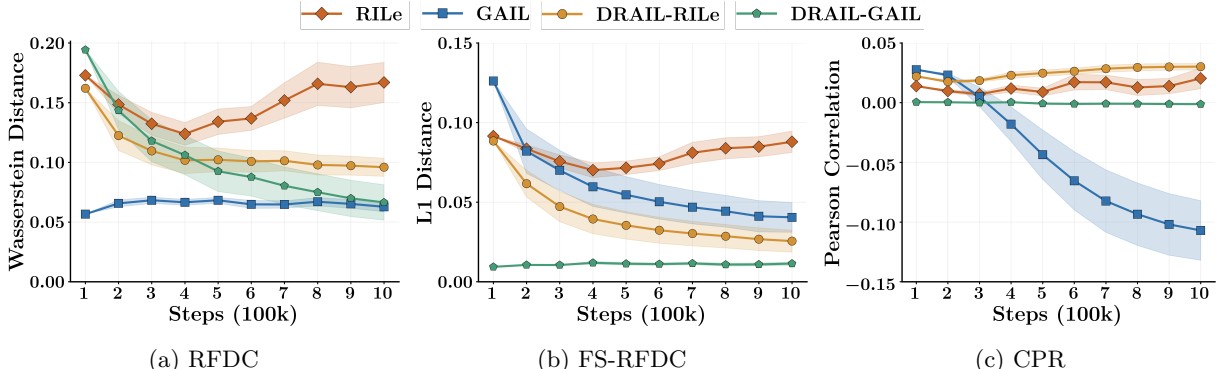

Figure 4: **Dynamics of Reward Functions**. **(a) Reward Function Distribution Change (RFDC):** Wasserstein distance between reward function distributions. **(b) Fixed-State Reward Function Distribution Change (FS-RFDC):** Mean absolute deviation of reward values for a fixed set of expert states. **(c) Correlation between Performance and Reward (CPR):** Pearson correlation between changes in the reward function and changes in the student's performance.

## 5.1 Ablation Studies

### 5.1.1 Reward Function Evaluation

To qualitatively evaluate how RILe's reward-learning strategy differs from AI(R)L baselines, we compare them in a maze environment. In this environment, the agent must navigate from a fixed start to a goal while avoiding static obstacles; we use a single expert demonstration.

Figure 3 shows how each method's learned reward function evolves during training. For RILe, we plot the trainer's learned reward function. For GAIL and AIRL, we visualize the discriminator outputs. The columns represent reward landscapes at 25%, 50%, 75%, and 100% of the total training process, and each subplot overlays the student's trajectory from the previous epoch.

RILe's reward function dynamically adapts to the student's current policy, providing guidance that encourage suboptimal actions which eventually lead the student closer to the expert trajectory. By contrast, GAIL and AIRL's reward functions remain relatively static. Specifically, the first column in Figure 3 shows RILe's trainer encouraging exploration toward the bottom-right of the maze, which is initially suboptimal but helpful in the long run. As the student learns to reach the lower part of the maze, RILe shifts high-reward regions toward the top-left (second column), again encouraging incremental progress. The third column illustrates how RILe boosts rewards near the goal while still maintaining some incentive around top-left areas to keep the agent from getting stuck.

Overall, as a qualitatively observed benefit, RILe's evolving reward function serves as a curriculum that promotes gradual improvement toward expert-like performance by encouraging exploration. This dynamic reward adaptation gets important in higher-dimensional tasks as we show in Section 5.3.

### 5.1.2 Reward Function Dynamics

To qualitatively evaluate how RILe's reward function evolves during the training, we compare RILe with GAIL, DRAIL-GAIL, and DRAIL-RILe in a high-dimensional robotic control scenario (learning to walk with UnitreeH1 robot).

We introduce three metrics (see Appendix D.2 for details): (1) RFDC (Reward Function Distribution Change): Wasserstein distance between reward distributions over consecutive training intervals, capturing overall shifts in reward space, (2) FS-RFDC (Fixed-State Reward Function Distribution Change): Mean absolute deviation of reward values at a fixed set of expert states over time, (3) CPR (Correlation between Performance and Reward): Pearson correlation between changes in the reward function and changes in the student's performance.

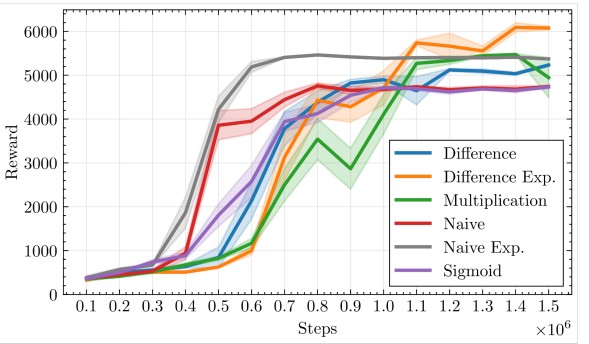 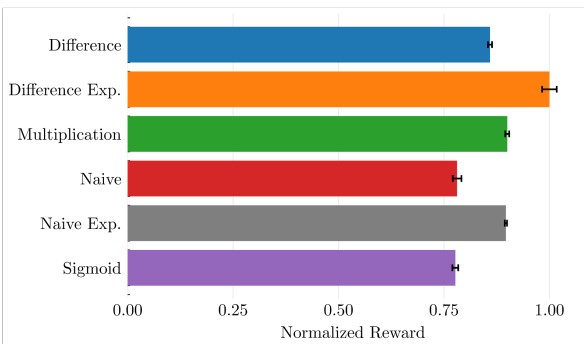

Figure 5: **Trainer-Discriminator Relation:** Comparison of different trainer reward functions, each defining a different relationship between the trainer's action and the discriminator's output. The student's return curves on the left show how performance evolves, and the normalized final performance on the right presents a clear comparison between reward designs. Exponential naive converges faster but plateaus at a lower final reward, whereas exponential difference yields the highest performance.

**Adaptability of the Learned Reward Function**  We compare the adaptability of the reward functions learned by RILe and AIL. Fig. 4a presents changes in reward distributions over 10,000 consecutive steps. RILe exhibits the highest adaptability in its reward function, aligning with our goal of having the reward function adapt based on the student's learning stage. The advanced discriminator in DRAIL reduces the need for drastic reward function changes, yet RILe remains more adaptive than GAIL. Since the changing student policy indirectly affects RFDC, we also show changes in reward values for the fixed set of states in Fig. 4b. Again, RILe's reward function is the most adaptive among all methods. Higher adaptability of RILe ensures that the reward signal remains aligned with incremental performance improvements, enabling the student to receive more timely and effective guidance throughout training.

**Correlation between the Learned Reward and the Student Performance**  We evaluate how changes in the reward function correlate with improvements in student performance. To this end, Fig. 4c presents the Pearson correlation between student's performance and reward updates. DRAIL-RILe achieves the highest positive correlation, indicating that it learns the most effective rewards for improving student performance. RILe ranks second, demonstrating that the trainer agent effectively helps the student achieve better scores even with the help of a naive-discriminator. In contrast, GAIL's correlation starts positive but soon turns negative and remains so throughout training. We hypothesize that this occurs because the discriminator in GAIL tends to saturate as training progresses. While the discriminator's reward signal effectively guides learning early on, its increasingly static nature at later stages fails to capture subtle performance improvements, leading to a negative correlation.

### 5.1.3  Trainer-Discriminator Relation

We investigate how the interaction between the trainer agent and the discriminator affects RILe's performance by comparing different trainer reward functions. Each reward function defines a different relationship between the trainer's action $a^T$ and the discriminator's output $D_\phi(s^T)$. We consider following reward functions: (a) **Difference** ($R^T = -|\upsilon(D_\phi(s^T)) - a^T|$), (b) **Exponential Difference** (default in RILe): $R^T = e^{-|\upsilon(D_\phi(s^T)) - a^T|}$, (c) **Multiplication** ($R^T = \upsilon(D_\phi(s^T))a^T$), (d) **Naive** ($R^T = D_\phi(s^T)$), (e) **Exponential Naive** ($R^T = e^{1-D_\phi(s^T)}$) and (f) **Sigmoid** ($(R^T = D_\phi(s^T)\sigma(a^T))$), where $\upsilon(x) = 2x - 1$ and $\sigma(x) = \frac{1}{1+e^{-x}}$.

Figure 5 presents reward curves and normalized rewards, where all rewards are normalized according to the maximum mean achieved test reward. While the exponential naive reward function offers the fastest convergence, the exponential difference reward offers the best performance. Therefore, we use exponential difference reward as the default reward function in RILe.

### 5.1.4 Robustness to Noise and Covariate Shift

We evaluate RILe's robustness to noisy expert demonstrations and environmental covariate shift. First, to evaluate the noise robustness of RILe In the MuJoCo Humanoid-v2 environment, we inject a zero-mean Gaussian noise (varying $\Sigma$) into either expert actions or states. We use GAIL (Ho & Ermon, 2016), AIRL (Fu et al., 2018), RIL-Co (Tangkaratt et al., 2021), IC-GAIL (Wu et al., 2019), and IQ-Learn (Garg et al., 2021) as baselines. Table 1 shows that RILe consistently outperforms baselines across all noise levels. Notably, it maintains high performance even under heavy noise ($\Sigma = 0.5$).

Second, inspired by Xu et al. (2022), we evaluate the stability of the reward functions learned by RILe and AIRL. First, we train both models in a clean environment. Then, we freeze the learned reward functions and train new student agents in environments where Gaussian noise is injected into their actions (covariate shift). Table 2 shows that the reward function learned by RILe demonstrates superior robustness to covariate shift, maintaining high performance even under increased noise levels.

Table 1: Robustness to different noise levels in the expert demonstrations in MuJoCo Humanoid-v2 environment.

|  |  | RILe | GAIL | AIRL | RIL-Co | IC-GAIL | IQ |
|---|---|---|---|---|---|---|---|
| Noise Free | $\Sigma = 0$ | **5928** | 5709 | 5623 | 576 | 610 | 327 |
| Action | $\Sigma = 0.2$ | **5280** | **5275** | 4869 | 491 | 601 | 192 |
|  | $\Sigma = 0.5$ | **5154** | 902 | 4589 | 493 | 568 | 153 |
| State | $\Sigma = 0.2$ | **5350** | 5147 | 4898 | 505 | 590 | 243 |
|  | $\Sigma = 0.5$ | **5205** | 917 | 4780 | 501 | 591 | 277 |

Table 2: Robustness to covariate shifts in environment.

|  | RILe | AIRL |
|---|---|---|
| No Noise | **5928** | 5623 |
| Mild $\Sigma = 0.2$ | **5201** | 5005 |
| High $\Sigma = 0.5$ | **5196** | 4967 |

### 5.1.5 Explicit Usage of Expert Data inside RILe

To assess the impact of using expert data explicitly inside RILe, we experiment in the MuJoCo Humanoid environment using a single expert trajectory from (Garg et al., 2021). We vary the proportion of expert data in the replay buffers from 0% to 100% (e.g., 25% indicates one quarter of each buffer is expert data; see Appendix D.5).

Figure 6 shows that while more expert data in both the trainer's and the student's replay buffers accelerates RILe's convergence, it reduces final performance. At 100% expert data, the student's performance drops markedly. This indicates that excessive expert data hampers the trainer's real-time adaptation, disrupting RILe's context-sensitive reward customization.

We compare RILe with IQ-Learn and BC, both of which rely heavily on expert data. Even with substantial amounts of expert data, RILe still performs better than baselines, indicating that RILe's adaptive reward-shaping provides a crucial edge over those methods.

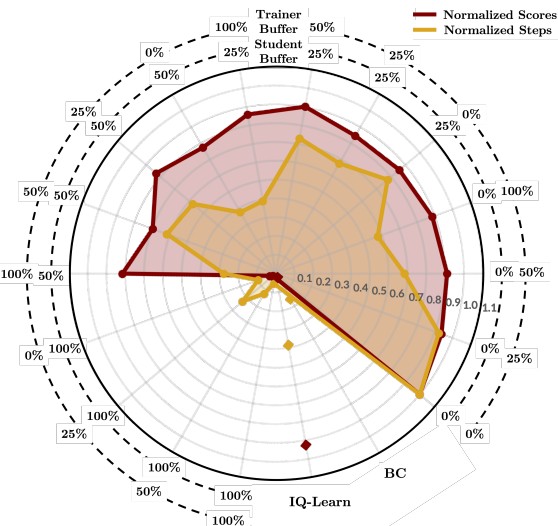

Figure 6: **Explicit Usage of Expert Data.** Red and yellow markers show normalized scores and steps. Expert data usage speeds the training of RILe but reduce final performance.

## 5.2 Learning from Expert Demonstrations

We evaluate how RILe's performance compares to baselines in a widely used learning from demonstration benchmark (Todorov et al., 2012; Brockman et al., 2016), where perfect expert state-action pairs are available.

Table 3: Test results on four MuJoCo tasks.

|  | RILe | GAIL | AIRL | IQ |
|---|---|---|---|---|
| Humanoid | **5928** | 5709 | 5623 | 327 |
| Walker2d | 4435 | **4906** | 4823 | 270 |
| Hopper | **3417** | 3361 | 3014 | 310 |
| HalfCheetah | **5205** | 4173 | 3991 | 755 |

Table 3 shows RILe consistently achieves competitive or superior performance compared to AIL/AIRL/IRL methods. In particular, on high-dimensional tasks (e.g. Humanoid), RILe's performance advantage is evident, underscoring the effectiveness of its adaptive reward function.

### 5.3 Robotic Continuous Control from Motion Capture Data

We test RILe in high-dimensional robotic locomotion tasks (Al-Hafez et al., 2023),for various robotic bodies. The agent must imitate motion-capture data, which is inherently noisy and only consists states without action information. This benchmark is especially demanding due to its complexity and dimensionality.

Table 4: Test results on seven LocoMujoco tasks.

|  |  | RILe | GAIL | AIRL | IQ | BCO | GAIfO | DRAIL GAIL | DRAIL RILe | Expert |
|---|---|---|---|---|---|---|---|---|---|---|
| Walk | Atlas | 870.6 | 792.7 | 300.5 | 30.9 | 21.0 | 834.2 | 834.4 | **899.1** | 1000 |
|  | Talos | 842.5 | 442.3 | 102.1 | 4.5 | 11.9 | 710.0 | 787.7 | **896.6** | 1000 |
|  | UnitreeH1 | 966.2 | 950.2 | 568.1 | 8.8 | 34.8 | 526.8 | 940.8 | **995.8** | 1000 |
|  | Humanoid | **831.3** | 181.4 | 80.1 | 4.5 | 3.5 | 706.5 | 814.6 | 527.6 | 1000 |
| Carry | Atlas | **850.8** | 669.3 | 256.4 | 36.8 | 20.3 | 810.1 | 516.6 | 317.1 | 1000 |
|  | Talos | 220.1 | 186.3 | 134.2 | 10.5 | 10.3 | 212.5 | 836.7 | **840.5** | 1000 |
|  | UnitreeH1 | 788.3 | 634.6 | 130.5 | 14.4 | 21.1 | 604.5 | 796.7 | **909.5** | 1000 |

Table 4 presents results for seven LocoMujoco tasks across different test seeds (see Appendix D.3 for details). Overall, these results underscore that RILe outperforms the AIL/IL/IRL baselines, and benefits even more from the enhancements provided by the DRAIL variants, achieving performance levels close to the expert. The performance variations across tasks indicate that although RILe's adaptive reward function is highly effective, task-specific factors also play a role. This overall performance aligns with our claim that an adaptive reward function is crucial for mastering complex, high-dimensional behaviors

## 6 Discussion

As our experiments demonstrate, RILe consistently outperforms baseline models across various tasks, thanks to its adaptive learning approach, where the trainer agent continuously adjusts the reward based on the student's current learning stage.

Our maze experiments illustrates how the trainer agent tailors its rewards. By encouraging actions that might seem suboptimal for immediate imitation but advantageous for long-term learning, RILe establishes a curriculum that ultimately boosts performance. This adaptive strategy helps RILe achieve superior results in our continuous control experiments, where reward shaping becomes especially critical in high-dimensional settings.

Nonetheless, policy stability remains challenging with dynamically evolving rewards. Freezing the trainer (see Appendix C) stabilizes learning but halts further adaptation, and the discriminator itself tends to overfit quickly. Future work may explore cooperative multi-agent RL to support continual adaptation, and consider discriminator-less formulations for reward learning. Furthermore, the nature of the trainer's reward learning suggests promising research directions. Since the trainer agent uses its own long-horizon value function to distill a reward-generating policy, the trainer acts more like a learned value estimate than a purely local reward. Exploring how to explicitly design trainer agents to provide different forms of guidance, including value-based signals, uncertainty estimates, and hierarchical curricula, is also a promising future direction.

Despite these challenges, RILe shows that *cooperatively learning* the policy and the reward function can offer significant advantages over static or iteratively updated methods. By providing dynamic and tailored rewards, RILe effectively guides the student through complex tasks. We believe this opens up new possibilities for responsive and adaptive learning frameworks in imitation learning and beyond.

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

## A   POMDP of the Trainer

Partially Observable Markov Decision Process (POMDP) of the trainer is defined as $\text{POMDP}_T = (S_T, A_T, \Omega_T, T_T, O_T, R_T, \gamma)$. Here, $T_T = \{P(. \mid f^T, a^T)\}$ is the transition dynamics where $P(. \mid f^T, a^T)$ is the state distribution upon taking action $a \in A_T$ in state $f \in S_T$. The transition function incorporates the student's policy $\pi_S$, which evolves in response to the rewards provided, reflecting the hidden dynamics due to the unobserved $\pi_S$. The observation function $O_T = \{P(s^T \mid f^T, a^T)\}$ defines the probability of observing $s^T \in \Omega_T$ given the state $(f^T, a^T)$. The trainer deterministically observes the student's state-action pair, so $P(s^T = (s^S, a^S) \mid f^T, a^T) = 1$, where $f^T = (s^S, a^S, \pi_S)$.

## B   Justification of RILe

**Assumptions:**

- The cross-entropy loss is convex with respect to the discriminator's outputs, and the discriminator function, $D_\phi(s, a)$, is sufficiently expressive with adequate model capacity.

- For the trainer's and student's policy functions $(\pi^{\theta_T})$ and $(\pi^{\theta_S})$, and the Q-functions $(Q^{\theta_S})$, each is Lipschitz continuous with respect to its parameters with constants $(L_{\theta_T}), (L_{\theta_S}), and (L_Q)$, respectively. This means for all $(s, a)$ and for any pair of parameter settings $(\theta, \theta') : [|\pi^\theta(s, a) - \pi^{\theta'}(s, a)| \leq L_\theta |\theta - \theta'|,][|Q^\theta(s, a) - Q^{\theta'}(s, a)| \leq L_Q |\theta - \theta'|.]$

To prove that the student agent can learn expert-like behavior, we need to show that the trainer agent learns to give higher rewards to student experiences that match with the expert state-action pair distribution, as this would enable a student policy to eventually mimic expert behavior.

### B.1   Lemma 1:

Given the discriminator $D_\phi$, the trainer agent optimizes its policy $\pi^{\theta_T}$ via policy gradients to provide rewards that guide the student agent to match expert's state-action distributions.

**Proof for Lemma 1** The student agent, $\pi_S(a_t^S | s_t^S)$, interacts with the environment and generates state-action pairs as $(s_t^S, a_t^S)$. The trainer agent observes these pairs and provides a reward $r_t^S = a_t^T = \pi_T(a_t^T | (s_t^S, a_t^S))$ to the student, where $a_t^T \in [-1, 1]$ is the trainer's action. We have $D_\phi : \mathcal{S} \times \mathcal{A} \to [0, 1]$ as the discriminator, parameterized by $\phi$, which outputs the likelihood that a given state-action pair $(s, a)$ originates from the expert, as opposed to the student.

The trainer's reward at timestep $t$ is:

$$r_t^T = e^{-|\upsilon(D_\phi(s_t^T)) - a_t^T|} \tag{11}$$

where $s_t^T = (s_t^S, a_t^S)$ is the trainer's observation, $D_\phi(s_t^T)$ is the discrimantor output that estimates the likelihood that $s_t^T$ comes from the expert data, and $\upsilon(D) = 2D - 1$ is a scaling function that maps discriminator's output to the range $[-1, 1]$.

The trainer maximizes the expected cumulative reward:

$$J_T(\pi_T) = \mathbb{E}_{\pi_T, \pi_S} \left[ \sum_{t=0}^{\infty} \gamma^t \left( r_t^T + \alpha H(\pi_T(\cdot | s_t^T)) \right) \right] \tag{12}$$

where $\gamma \in [0, 1)$ is the discount factor and $\alpha$ is the entropy weight. In other words, trainer aims to find the policy that maximizes $J_T(\pi_T)$: $\pi^{*T} = \arg\max_{\pi^T} J_T(\pi_T)$.

From the policy gradient theorem, the gradient of the trainer's objective with respect to the policy parameters, $\theta_T$, is:

$$\nabla_{\theta_T} J_T(\pi_T) = \mathbb{E}_{\pi_T, \pi_S} \left[ \nabla_{\theta_T} \log \pi_T(a_t^T | s_t^T) \left( Q_T(s_t^T, a_t^T) - \alpha \log \pi_T(a_t^T | s_t^T) \right) \right] \tag{13}$$

where $Q_T(s_t^T, a_t^T)$ is the soft action-value function of the trainer. The soft action-value function, $Q_T(s_t^T, a_t^T)$, and the soft value function, $V_T(s_t^T)$ is defined by Bellman equation as:

$$Q_T(s_t^T, a_t^T) = r_t^T + \gamma \mathbb{E}_{s_{t+1}^T} \left[ V_T(s_{t+1}^T) \right] \tag{14}$$

$$V_T(s_t^T) = \mathbb{E}_{a_t^T \sim \pi_T} \left[ Q_T(s_t^T, a_t^T)) - \alpha \log(\pi_T(\cdot | s_t^T)) \right] \tag{15}$$

The trainer aims to take an action $a_t^T$ that maximises the expected $Q_T(s_t^T, a_t^T)$ to satisfy Equation 13. Since $r_t^T$ depends directly on $D_\phi(s_t^T)$ and $a_t^T$, the trainer learns to select $a_t^T$ that maximizes $Q_T(s_t^T, a_t^T)$. The optimal policy and action depends heavily on $\gamma$. If the trainer agent is myopic ($\gamma_T = 0$), the future value term in Equation 14 disappears, and the soft Q-function becomes equal to the immediate reward: $Q(s_t^T, a_t^T) = r_t^T = e^{-|v(D_\phi(s_t^T)) - a_t^T|}$. Considering that $a_t^T \in [-1, 1]$, the immediate reward $r_t^T$ is maximized when $a_t^T$ matches $v(D_\phi(s_t^T))$. Therefore, in this myopic case, the optimal action $a_t^{*T}$ is $\alpha_t^{*T} = v(D_\phi(s_t^T)) = 2D_\phi(s_t^T) - 1$. Thus, in the myopic setting, the trainer's policy aims to select actions centered around a static transformation of the discriminator's output.

In the full RILe framework, $\gamma > 0$, and the trainer agent is long-horizoned. The Q-function is:

$$Q_{soft}(s_t^T, a_t^T) = e^{-|v(D_\phi(s_t^T)) - a_t^T|} + \gamma \mathbb{E}_{s_{t+1}^T} \left[ V(s_{t+1}^T) \right] \tag{16}$$

$$= e^{-|v(D_\phi(s_t^T)) - a_t^T|} + \gamma \mathbb{E}_{s_{t+1}^T} \left[ \mathbb{E}_{a_t^T \sim \pi_T} \left[ Q_T(s_{t+1}^T, a_{t+1}^T)) - \alpha \log(\pi_T(\cdot | s_{t+1}^T)) \right] \right] \tag{17}$$

Here, the optimal action $a_t^{*T}$ must consider long-horizon effects and balance maximizing the immediate reward with maximizing the expected future rewards. The distribution of the next state, $s_{t+1}^T = (s_{t+1}^S, a_{t+1}^S)$, depends on the student policy, which is guided by the reward $r_t^S = a_t^T$ provided by the trainer at the current step. As shown Appendix B.4, the optimal action is not a function only of the current discriminator output $D_\phi(s_t^T)$, but also of the student's current policy $\pi_S$ and the environment dynamics $T$.

All in all, the trainer agent learns a policy that considers immediate and future rewards from the discriminator. The trainer provides rewards that guide the student toward trajectories that are expected to yield high cumulative returns from the discriminator. Therefore the trainer learns to assign higher rewards to student behaviors that are more similar to expert behaviors, according to the discriminator.

## B.2  Lemma 2:

The discriminator $D_\phi$, parameterized by $\phi$ will converge to a function that estimates the probability of a state-action pair being generated by the expert policy, when trained on samples generated by both a student policy $\pi^{\theta_S}$ and an expert policy $\pi_E$.

**Proof for Lemma 2**: The discriminator's objective is to distinguish between state-action pairs generated by the expert and those generated by the student. The training objective for the discriminator is framed as a binary classification problem over expert demonstrations and student-generated trajectories. The discriminator's loss function $\mathcal{L}_D(\phi)$ is the binary cross-entropy loss, which is defined as:

$$L_D(\phi) = -\mathbb{E}_{(s,a) \sim p_E}[\log(D_\phi(s, a))] - \mathbb{E}_{(s,a) \sim p_{\pi_S}}[\log(1 - D_\phi(s, a))]. \tag{18}$$

where $p_E(s, a)$ is the state-action distribution of the expert policy, and $p_{\pi_S}(s, a)$ is the state-action distribution of the student agent. Considering that $x = (s, a)$, this loss can be rewritten as:

$$L_D(\phi) = -\int [p_E(s, a) \log D_\phi(s, a) + p_{\pi_S}(s, a) \log(1 - D_\phi(s, a))] \, ds \, da \tag{19}$$

$$L_D(\phi) = -\int [p_E(x) \log D_\phi(x) + p_{\pi_S}(x) \log(1 - D_\phi(x))] \, dx. \tag{20}$$

As presented in Goodfellow et al. (2014), the optimal discriminator that minimizes this loss, $D_\phi^*$, is:

$$D_\phi^*(x) = \frac{p_E(x)}{p_E(x) + p_{\pi_S}(x)}, \tag{21}$$

$$D_\phi^*(s,a) = \frac{p_E(s,a)}{p_E(s,a) + p_{\pi_S}(s,a)}. \tag{22}$$

This shows that the optimal discriminator estimates the probability that a state-action pair comes from the expert policy, normalized by the total probability from both expert and student policies.

## B.3    Motivation for the Trainer Agent

The fundamental limitation of adversarial learning approaches lies in the nature of their objective functions. The reward signal in AIL is a direct byproduct of a *myopic*, binary classification objective aimed at instantaneously separating expert and student data. The optimal discriminator converges to a quasistatic function of the expert and policy densities, $D^*(s,a) = p_E(s,a)/(p_E(s,a) + p_{\pi_S}(s,a))$ (see Appendix B.2). A reward derived statically from this function is also *myopic*, tends to saturate once the discriminator becomes confident, providing coarse, binary-like feedback that is often insufficient for guiding an agent through complex, high-dimensional tasks. In contrast, RILe's trainer is a fully separate reinforcement learning agent whose objective is to maximize a long-horizon, discounted sum of future discriminator rewards (Eq. 10). The trainer learns a reward-generating *policy*, not just a static function of the discriminator. This allows it to provide a seemingly suboptimal reward at the current step if its value function, $Q_T$, predicts this will lead to higher discriminator scores in the long run.

The RL-based architecture also allows the trainer to *explore reward strategies*. Because the trainer $\pi_T$ is an RL agent, its reward-giving action $a^T$ is not tied to the discriminator's instantaneous judgment. It is incentivized to explore different actions (i.e., different reward values for the student) for the same state, a process encouraged by an entropy regularization term, $H(\pi_T)$, in its objective. This allows the trainer to gradually learn to steer the student into states that yield higher long-term rewards, even if the discriminator's immediate reward is low. The result is a dynamically changing reward landscape that emphasize different subgoals as the student improves, a curriculum effect that a static transformation of $D_\phi$ fails to replicate.

RILe establishes a two-level learning dynamic rather than a fully adversarial setting. The core interaction between the student and the trainer is cooperative. The student works to maximize the rewards provided by the trainer, while the trainer learns to provide rewards that effectively guide the student toward expert-like behavior. Their goals are aligned: for the student to successfully imitate the expert and fool the discriminator. This cooperative student-trainer pair then operates within a broader adversarial game, leveraging the feedback from the discriminator which remains in a competitive relationship with the student's generated trajectories.

## B.4    Theoretical Justification of the Trainer Agent

The reward policy $\pi_T$ learned by the RILe trainer agent is fundamentally distinct from any static transformation $g(D_\phi)$ of the discriminator's output, except in the degenerate case where the trainer's learning objective is myopic (i.e., its discount factor $\gamma_T = 0$).

**Definition (Myopic Reward)**: A reward $r(s_t, a_t)$ is *myopic* if it depends only on the current discriminator output $D_\phi(s_t, a_t)$ and *not* on any future transitions or on the policy's evolution.

**Analysis:** In frameworks like GAIL or a hypothetical variant, the reward given to the student is a fixed, or *static*, transformation, $g(D_\phi)$, of the discriminator's output:

$$r_g(s_t, a_t) = g(D_\phi(s_t, a_t)) \tag{23}$$

By definition, this reward signal is myopic. The key characteristic is that its value depends only on the instantaneous output of the discriminator and is independent of future consequences and environment dynamics.

In contrast, RILe's student reward is the action of the trainer agent:

$$r_S(s_t, a_t) = a_t^T \tag{24}$$

where $a_t^T \sim \pi_T(\cdot | s_t^T)$. The trainer is a full reinforcement learning agent, and the trainer's policy $\pi_T$ is optimized to maximize its own long-horizon objective:

$$\pi_T^* = \arg\max_{\pi_T} \mathbb{E}_{\substack{s_t^T \sim \pi_S \\ a_t^T \sim \pi_T}} \left[ \sum_{t=0}^{\infty} \gamma_T^t \left( R_t^T + \alpha H(\pi_T(\cdot \mid s_t^T)) \right) \right]. \tag{25}$$

where the crucial element is the discount factor $\gamma_T > 0$.

The core of difference lies in the definition of the trainer's action-value function, $Q_T^*(s^T, a^T)$, which the policy $\pi_T^*$ maximizes. According to the Bellman equation, $Q_T^*$ is defined recursively:

$$Q_T^*(s_t^T, a_t^T) = R_t^T + \gamma_T \mathbb{E}_{s_{t+1}^T \sim P(\cdot | s_t^T, a_t^T)} \left[ V_T^*(s_{t+1}^T) \right]. \tag{26}$$

The key distinction lies in the second term:

$$\gamma_T \mathbb{E}[V_T^*(s_{t+1}^T)] = \gamma_T \mathbb{E}_{s_{t+1}^S \sim T(\cdot | s_t^S, a_t^S), a_{t+1}^S \sim \pi_S(\cdot | s_{t+1}^S)} [V_T^*(s_{t+1}^T)] \tag{27}$$

This term represents the discounted value of all future states and inextricably links the trainer's current action to its long-term consequences. The distribution of the next trainer state, $s_{t+1}^T$, is a function of the environment's dynamics, $T$, and the student's current policy, $\pi_S$. Consequently, the optimal trainer action $a_t^{*T} = \arg\max_{a_t^T} Q_T^*(s_t^T, a_t^T)$ has far richer dependencies than a static function:

$$a_t^{*T} = f(D_\phi(s_t, a_t), \gamma_T, T, \pi_S)$$

Because the trainer's reward signal is dependent on its discount factor $\gamma_T$, environment dynamics $T$, and the student's policy $\pi_S$, it cannot be reduced to a static transformation $g(D_\phi)$, which lacks these dependencies. The trainer learns a strategic, forward-looking teaching policy rather than executing a reactive, myopic mapping.

The only scenario where this distinction vanishes is the degenerate case where $\gamma_T = 0$. If the trainer is myopic, the future-looking term in Equation 26 disappears. The objective collapses to maximizing the immediate reward $R_t^T$, making the trainer's action a deterministic function of $D_\phi$ and thus functionally equivalent to a static transformation.

Finally, the trainer's objective is also optimized with an entropy regularization term, $\alpha H(\pi_T)$, which forces the policy to be *stochastic*. A policy that outputs a distribution over rewards cannot be equivalent to a *deterministic* function like $g(D_\phi)$, providing a second, independent reason for their non-equivalence.

**Proposition 1:** Let $g : [0, 1] \to [-1, 1]$ be any deterministic function. There exists an MDP, student policy $\pi_S$, and corresponding discriminator $D_\phi$ for which the optimal trainer action $a_T^*(s, a)$ differs between two contexts despite $D_\phi(s, a)$ being identical. Hence no static reward $(r_g(s, a) = g(D_\phi(s, a)))$ can match the long-horizon shaping of $\pi_T^*$.

**Proof for Proposition 1** We construct a simple 1-step MDP. Let the state space be $\mathcal{S} = \{s_0, s_1, s_2\}$, where $s_0$ is the initial state and $s_1, s_2$ are terminal states. From $s_0$, the student takes action $a_0$. The expert demonstration is the trajectory $\tau_E = (s_0, a_0, s_2)$, establishing $s_2$ as the desirable outcome. We assume the discriminator's output for the initial state-action pair as $D_\phi(s_0, a_0) = d_0$. Within the RILe framework, trajectories ending in the expert state $s_2$ yield higher long-term cumulative rewards for the trainer than those ending in $s_1$. We thus define the trainer's terminal state values as $V_T(s_1) = V_{low}$ and $V_T(s_2) = V_{high}$, where $V_{high} > V_{low}$.

The trainer's action $a_T$ at $(s_0, a_0)$ becomes the student's reward, which influences the student's policy and thus the state transition probabilities. We model the probability of reaching the desirable state $s_2$ as a function of $a_T$ using the sigmoid function $\sigma(x) = (1 + e^{-x})^{-1}$, such that $P(s' = s_2 | s_0, a_T) = \sigma(\beta a_T)$. The

parameter $\beta$ models the student's responsiveness. We analyze two contexts representing different student learning stages. In Context C (Eager Student), the student responds positively to reward, so we set $\beta = k$ for some $k > 0$. In Context C' (Naive Student), the student responds perversely to reward because of heavy exploration, so we set $\beta = -k$.

The trainer chooses $a_T$ at $s_0$ to maximize its Q-value, which is the sum of the immediate reward and the discounted expected future value:

$$Q_T(s_0, a_T) = e^{-|v(d_0) - a_T|} + \gamma_T \left[ P(s_2|a_T)V_{high} + (1 - P(s_2|a_T))V_{low} \right]$$

where $v(d_0) = 2d_0 - 1$. In Context C, $P_C(s_2|a_T) = \sigma(ka_T)$. To maximize $Q_T$, the trainer is incentivized to choose a high $a_T$, as this maximizes both the immediate and expected future reward terms. In Context C', $P_{C'}(s_2|a_T) = \sigma(-ka_T)$. Here, the trainer faces a trade-off: a high $a_T$ maximizes the immediate reward but minimizes the future reward. To maximize the total Q-value, the trainer must choose a low $a_T$ to steer the student to $s_2$. Since the optimal action $a_T^*$ differs between contexts for the same input $d_0$, no static function $g(d_0)$ can replicate this behavior.

## C    Training Strategies

The introduction of the trainer agent into the AIL framework introduces instabilities that can hinder the learning process. To address these challenges, we employ three strategies.

**Freezing the Trainer Agent Midway:** Continuing to train the trainer agent throughout the entire process leads to overfitting on minor fluctuations in the student's behavior. This overfitting causes the trainer to assign inappropriate negative rewards, which diverts the student away from expert behavior—especially since the student agent may fail to interpret these subtle nuances correctly in the later stages of training. To prevent this, we freeze the trainer agent (and the discriminator) once its critic network within the actor-critic framework converges during the training process.

**Utilizing a Smaller Buffer for the Trainer Agent:** We employ distinct replay buffer sizes for the student and trainer agents. We use larger buffer for the student compared to the trainer, as detailed in our hyperparameter configurations (see Appendix G). This strategy ensures the trainer primarily learns from the student's recent interactions, allowing it to adapt its reward function more rapidly to the evolving student policy instead of optimizing based on potentially outdated historical data. This increased responsiveness provides more relevant, timely feedback to the student, which we found empirically contributes to more stable and effective co-adaptation within the RILe framework across different tasks.

**Increasing the Student Agent's Exploration:** We increase the exploration rate of the student agent compared to standard AIL methods. We implement an epsilon-greedy strategy within the actor-critic framework, allowing the student to occasionally take random actions. This increased exploration enables the student to visit a wider range of state-action pairs. Consequently, the trainer agent receives diverse input, helping it learn a more effective reward function. This diversity is crucial for the trainer to observe the outcomes of various actions and to guide the student more effectively toward expert behavior.

## D    Experimental Settings

### D.1    Evolving Reward Function

We use single expert demonstration in this experiment. For RILe, we plot the reward function learned by the trainer. For GAIL, we visualize the discriminator output, and for AIRL, the reward term under the discriminator.

The 2D axes in the maze environment (Fig. 3) represent the state space $s^T = (s^S, a^S)$ of the trainer agent. To visualize the reward $R(s^T)$, for each student state $s^S$, we calculate reward for a fixed set of actions: $\mathcal{A} = \{(v_x, v_y) : v_x, v_y \in \{-1, -0.5, 0, 0.5, 1\}\}$. For each $s^S$ (for each x,y coordinate in the maze), we compute reward outputs for 25 actions, resulting in 5×5 slice of the reward surface. This provides a landscape of the rewards the agent could expect to receive across the maze. Every episode is initialized at

the same starting point, and the training completion percentage refers to the fraction of total training steps completed.

## D.2 Reward Function Dynamics

In this experiment, we select the student agent's hyperparameters to be identical to those used in GAIL, ensuring that the only difference between the agents is the reward function. Therefore, we use the best hyperparameters identified for GAIL, applied to both GAIL and RILe, from our hyperparameter sweeps presented in Appendix G.

**RFDC:** We calculate the Wasserstein distance between reward distributions over consecutive 10,000-step training intervals, denoted as times $t$ and $t + 10,000$. This metric quantifies how much the overall reward distribution shifts over time. Changes in reward distributions depend both on the reward function and the student policy updates. Since we use the same student agent with the same hyperparameters, higher RFDC values still indicate that the reward function is adapting more dynamically in response to the student's learning progress.

**FS-RFDC:** We compute the mean absolute deviation of rewards between consecutive 10,000-step training intervals for a fixed set of states derived from expert data. As the fixed set, we use all the states in the expert data. Since the states used for calculating rewards are fixed, changes in this value purely depend on the reward function updates. This metric assesses how the reward values for specific states change over time.

**CPR:** We evaluate how changes in the reward function correlate with improvements in student performance. We store rewards from both the learned reward function and the environment-defined rewards in separate buffers. In other words, we collect samples from two reward functions: the learned reward function and the environment-defined reward function. The environment rewards consider the agent's velocity and stability. Every 10,000 steps, we calculate the Pearson correlation between these rewards and empty the buffers. This metric evaluates whether increases in the learned rewards relate to performance enhancements.

## D.3 Motion-Capture Data Imitation for Robotic Continuous Control

During training, we use 8 different random seeds and 8 distinct initial positions for the robot. The validation setting mirrors the training conditions: we sample initial positions from the same set of 8 possibilities and use the same random seeds. In this setting, the student agent selects actions deterministically, allowing us to assess its performance under familiar conditions.

For the test setting, we evaluate the policy's ability to generalize to new, unseen scenarios. We modify the initial positions of the robot by randomly initializing it in stable configurations not included in the fixed set used during training. Additionally, we use different random seeds from those in training, introducing new random variations that affect the environment's dynamics during state transitions. This setup enables us to assess how well the learned policy performs when faced with novel initial conditions.

## D.4 Learning from Demonstrations

Each method is trained using 25 expert trajectories provided in the IQ-Learn paper Garg et al. (2021). We use single seed for the training, and after the training, run experiments with 10 different random seeds and report the mean and standard deviation of the results.

## D.5 Impact of Expert Data on Trainer-Student Dynamics

In this experiment, both seeds and initial positions in the test setting are different from the training one, and we report values from the test setting.

For every percentage of the expert-data in buffers, we continue trainings of both the trainer agent and the student agent of RILe. For instance, in 100% expert data in the trainer's buffer case, both the student and the discriminator are trained normally using samples from the student agent. However, we didn't include student's state-action pairs to the trainer's buffer, instead, we filled that buffer with a batch of expert data,

and updated the trainer regularly using this modified buffer. Similarly, in 100% expert data in the student's buffer case , we trained the trainer agent and the discriminator normally, using samples from the student. However, student's state-actions pairs are not included in the student's buffer, and student agent is updated just by using expert state-action pairs, using rewards coming from the trainer agent for these expert pairs.

Regarding the normalizations, we trained Behavioral Cloning (BC) and RILe across various data leakage levels, selecting the highest-scoring run (0% leakage RILe) as the baseline. Other scores and convergence steps are normalized by dividing by the score and convergence steps of the baseline (0% leakage RILe). For IQLearn, we used their reported numbers in their paper, as we couldn't replicate their results with their code and hyperparameters.

## E    Extended MuJoCo Results

We present MuJoCo results for the test setting, with standard errors, in Table 5.

Table 5: Test results on four MuJoCo tasks with standard errors.

|  | RILe | GAIL | AIRL | IQLearn | DRAIL |
|---|---|---|---|---|---|
| Humanoid-v2 | **5928 $\pm$ 188** | 5709 $\pm$ 63 | 5623 $\pm$ 252 | 327 $\pm$ 105 | 5755 $\pm$ 34 |
| Walker2d-v2 | 4435 $\pm$ 206 | **4906 $\pm$ 159** | 4823 $\pm$ 221 | 270 $\pm$ 43 | 4016 $\pm$ 127 |
| Hopper-v2 | **3417 $\pm$ 155** | 3361 $\pm$ 51 | 3014 $\pm$ 190 | 310 $\pm$ 47 | 1230 $\pm$ 73 |
| HalfCheetah-v2 | **5205 $\pm$ 31** | 4173 $\pm$ 94 | 3991 $\pm$ 126 | 755 $\pm$ 211 | 4133 $\pm$ 41 |

# F Extended LocoMujoco Results

We present LocoMujoco results for the validation setting and test setting, with standard errors, in Table 6 and 7, respectively.

Table 6: Validation results on seven LocoMujoco tasks.

| | | RILe | GAIL | AIRL | IQ | BCO | GAIfO | DRAIL GAIL | DRAIL RILe | Expert |
|---|---|---|---|---|---|---|---|---|---|---|
| Walk | Atlas | 895.4 ±25 | 918.6 ±133 | 356.0 ±68 | 32.1 ±4 | 28.7 ±4 | 831.6 ±41 | 741.3 ±46 | 773.9 ±13 | 1000 |
| | Talos | 884.7 ±8 | 675.5 ±105 | 103.4 ±22 | 7.2 ±2 | 19.9 ±4 | 718.8 ±16 | 963.7 ±48 | 949.4 ±54 | 1000 |
| | UnitreeH1 | 980.7 ±15 | 965.1 ±20 | 716.2 ±124 | 12.5 ±6 | 43.7 ±8.4 | 586.6 ±102 | 954.7 ±20 | 973.5 ±8 | 1000 |
| | Humanoid | 970.3 ±101 | 216.2 ±18 | 78.2 ±6 | 6.8 ±1 | 8.3 ±1 | 345.7 ±34 | 550.8 ±148 | 595.3 ±73 | 1000 |
| Carry | Atlas | 889.7 ±44 | 974.2 ±80 | 271.9 ±30 | 39.5 ±8 | 42.7 ±9 | 306.2 ±9 | 654.1 ±109 | 344.1 ±28 | 1000 |
| | Talos | 503.3 ±72 | 338.5 ±48 | 74.1 ±8 | 11.7 ±3 | 8.1 ±1 | 444.5 ±96 | 889.8 ±163 | 874.3 ±174 | 1000 |
| | UnitreeH1 | 850.6 ±80 | 637.4 ±90 | 140.9 ±21 | 12.3 ±2 | 30.2 ±5 | 503.6 ±55 | 620.8 ±60 | 878.1 ±46 | 1000 |

Table 7: Test results on seven LocoMujoco tasks.

| | | RILe | GAIL | AIRL | IQ | BCO | GAIfO | DRAIL GAIL | DRAIL RILe | Expert |
|---|---|---|---|---|---|---|---|---|---|---|
| Walk | Atlas | 870.6 ±13 | 792.7 ±105 | 300.5 ±74 | 30.9 ±10 | 21.0 ±3 | 803.1 ±68 | 834.4 ±23 | **899.1** ±17 | 1000 |
| | Talos | 842.5 ±24 | 442.3 ±76 | 102.1 ±17 | 4.5 ±3 | 11.9 ±1 | 687.2 ±44 | 787.7 ±11 | **896.6** ±12 | 1000 |
| | UnitreeH1 | 966.2 ±14 | 950.2 ±13 | 568.1 ±156 | 8.8 ±3 | 34.8 ±10 | 526.8 ±72 | 940.8 ±20 | **995.8** ±6 | 1000 |
| | Humanoid | **831.3** ±98 | 181.4 ±24 | 80.1 ±9 | 4.5 ±2 | 3.5 ±2 | 292.1 ±25 | 814.6 ±80 | 527.6 ±39 | 1000 |
| Carry | Atlas | **850.8** ±62 | 669.3 ±55 | 256.4 ±47 | 36.8 ±14 | 20.3 ±1 | 402.9 ±39 | 516.6 ±60 | 317.1 ±19 | 1000 |
| | Talos | 220.1 ±88 | 186.3 ±28 | 134.2 ±18 | 10.5 ±3 | 10.3 ±2 | 212.5 ±32 | 836.7 ±160 | **840.5** ±133 | 1000 |
| | UnitreeH1 | 788.3 ±71 | 634.6 ±45 | 130.5 ±22 | 14.4 ±2 | 21.1 ±6 | 504.5 ±30 | 796.7 ±131 | **909.5** ±9 | 1000 |

## G   Hyperparameters

We present hyperparameters in Table 8. We use SAC Haarnoja et al. (2018) as the RL architecture for both the student and trainer agents for RILe by default. For the Maze experiment (Section 5.1), we employ PPO Schulman et al. (2017). For DRAIL, we replaced the discriminators with the implementation provided by DRAIL and adopted their hyperparameters for the HandRotate task.

Our experiments revealed that RILe's performance is particularly sensitive to certain hyperparameters. We highlight three key observations:

- RILe is more sensitive to the hyperparameters of the discriminator compared to other methods. Specifically, increasing the discriminator's capacity or training speed, by using a larger network architecture or increasing the number of updates per iteration, adversely affects RILe's performance. A powerful discriminator tends to overfit quickly to the expert data, resulting in high confidence when distinguishing between expert and student behaviors. This poses challenges for the trainer agent, as the discriminator's feedback becomes less informative.

- Employing distinct replay buffer sizes, particularly a smaller buffer for the trainer agent compared to the student agent, offers better stability. This encourages the trainer to learn primarily from recent student interactions, allowing its reward function to consider recent advancements in the student's evolving policy, rather than optimizing based on potentially outdated historical data. This allows the trainer agent to be more responsive and provide more relevant, timely feedback.

- Enhancing the exploration rate of the student agent benefits RILe more than it does baseline methods. By encouraging the student to explore more, through strategies like higher entropy regularization or implementing an epsilon-greedy policy, the student visits a broader range of state-action pairs. This increased diversity provides the trainer agent with more varied data, enabling it to learn a more effective and robust reward function. The additional exploration helps the trainer to better capture the effects of different actions.

## H   Compute Resources

For the training of RILe and baselines, following computational sources are employed:
- AMD EPYC 7742 64-Core Processor
- 1 x Nvidia A100 GPU
- 32GB Memory

## I   Failure Modes and Limitations

While RILe demonstrates strong performance, its multi-agent framework introduces specific challenges and potential failure modes.

**Discriminator Overfitting:** The performance of RILe is highly dependent on a well-calibrated discriminator. A primary failure mode occurs if the discriminator learns too quickly or has excessive capacity relative to the task. When this happens, it can overfit to the expert data and provide near-binary, uninformative feedback to the trainer agent. This starves the trainer of the nuanced signal required to learn an effective, adaptive reward function, which is why RILe is particularly sensitive to the discriminator's hyperparameters.

**Co-adaptation Instability:** The dynamic co-evolution of the trainer and student, where the reward function is constantly changing, can lead to policy instability. If the trainer adapts its reward function too drastically in response to minor fluctuations in the student's policy, the student may struggle to learn consistently. To mitigate this, we introduced stabilizing strategies such as freezing the trainer mid-way through training once its critic converges and using a smaller replay buffer for the trainer to focus its learning on the student's most recent behaviors.

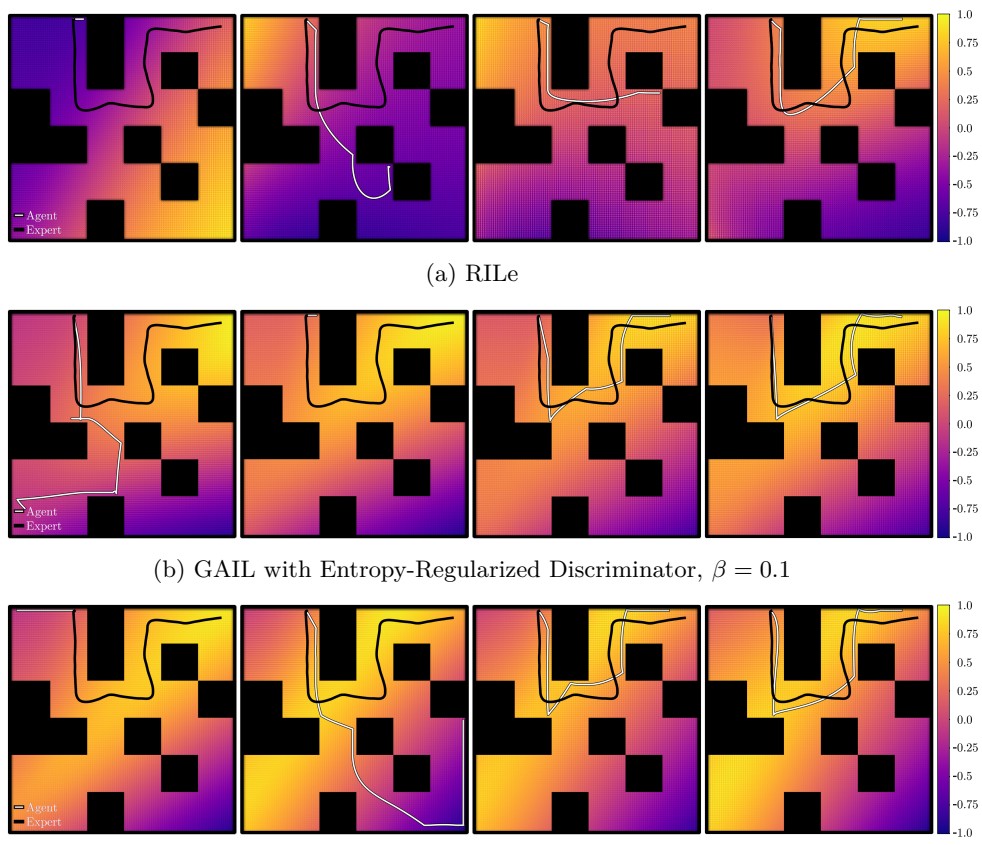

(a) RILe

(b) GAIL with Entropy-Regularized Discriminator, $\beta = 0.1$

(c) GAIL with Entropy-Regularized Discriminator, $\beta = 1.0$

Figure 7: Evolution of reward functions during training for (a) RILe, (b) GAIL with entropy-regularized discriminator ($\beta = 0.1$), and (c) GAIL with entropy-regularized discriminator ($\beta = 1.0$). RILe demonstrates a dynamic reward function that adapts with the student's progress. Although entropy regularization makes the reward landscape of GAIL smoother, still, the reward function stays relatively static during the training, without adaptation to the student's progress.

**Insufficient Student Exploration:** The trainer requires diverse data from the student to learn a robust and generalizable reward function. If the student's exploration is insufficient, the trainer may not observe the consequences of a wide enough range of behaviors, limiting its ability to provide effective guidance. Enhancing the student's exploration rate, for example through higher entropy regularization, is more beneficial for RILe than for baseline methods as it provides the trainer with more varied data.

## J    Entropy-Enhanced Discriminator in AIL

As a further ablation, we compare RILe with entropy-enhanced GAIL. We include entropy objective, $H(D)$, inside discriminator's objective in GAIL. Formally, we update the objective of the discriminator as:

$$\max_{\phi} \mathbb{E}_{\tau_E}[\log D_\phi(s,a)] + \mathbb{E}_\pi[\log (1 - D_\phi(s,a))] + \beta H(D_\phi). \tag{28}$$

We define the reward of the learning agent as $R(s,a) = v(D_\phi(s,a)) = 2D_\phi(s,a) - 1$.

We train entropy-regularized GAIL in the same maze environment that we use in Section 5.1.1. Figure 7 shows how the learned landscapes look, along with the student's trajectory from the previous epoch. Entropy regularization smoothens the GAIL's reward landscape created by the discriminator. However, still, the reward function remains relatively static during the training, without adapting to the student's progress.

Table 8: Hyperparameter Sweeps and Best Hyperparameters for LocoMujoco and Humanoid Experiments

| | Hyperparameters | RILe | GAIL | AIRL | IQ-Learn |
|---|---|---|---|---|---|
| **Discriminator** | Updates per Round | **1**, 2, 8 | **1**, 2, 8 | **1**, 2, 8 | - |
| | Batch Size | **32**, 64, 128 | **32**, 64, 128 | **32**, 64, 128 | - |
| | Buffer Size | 8192, **16384**, 1e5 | 8192, **16384**, 1e5 | 8192, **16384**, 1e5 | - |
| | Network | [512FC, 512FC] [256FC, 256FC] [**64FC, 64FC**] | [512FC, 512FC] [256FC, 256FC] [**64FC, 64FC**] | [512FC, 512FC] [256FC, 256FC] [**64FC, 64FC**] | - |
| | Gradient Penalty | 0.5, **1** | 0.5, **1** | 0.5, **1** | - |
| | Learning Rate | 3e-4, 1e-4, **3e-5**, 1e-5 | 3e-4, 1e-4, **3e-5**, 1e-5 | 3e-4, 1e-4, **3e-5**, 1e-5 | - |
| **Student** | Buffer Size | 1e5, **1e6** | 1e5, **1e6** | 1e5, **1e6** | 1e5, **1e6** |
| | Batch Size | 32, **256** | 32, **256** | 32, **256** | 32, **256** |
| | Network | [**256FC, 256FC**] | [**256FC, 256FC**] | [**256FC, 256FC**] | [**256FC, 256FC**] |
| | Activation Function | **ReLU**, Tanh | **ReLU**, Tanh | **ReLU**, Tanh | **ReLU**, Tanh |
| | Discount Factor ($\gamma$) | **0.99**, 0.97, 0.95 | **0.99**, 0.97, 0.95 | **0.99**, 0.97, 0.95 | **0.99**, 0.97, 0.95 |
| | Learning Rate | **3e-4**, 1e-4, 3e-5, 1e-5 | **3e-4**, 1e-4, 3e-5, 1e-5 | **3e-4**, 1e-4, 3e-5, 1e-5 | **3e-4**, 1e-4, 3e-5, 1e-5 |
| | Tau ($\tau$) | 0.05, **0.01**, 0.005 | 0.05, **0.01**, 0.005 | 0.05, **0.01**, 0.005 | 0.05, **0.01**, 0.005 |
| | Epsilon-greedy | **0**, 0.1, 0.2 | **0**, 0.1, 0.2 | **0**, 0.1, 0.2 | **0**, 0.1, 0.2 |
| | Entropy | **0.2**, 0.5, 1 | **0.2**, 0.5, 1 | **0.2**, 0.5, 1 | 0.05, 0.1, **0.2**, 0.5, 1 |
| **Trainer** | Buffer Size | 8192, **16384**, 1e5, 1e6 | - | - | - |
| | Batch Size | 32, **256** | - | - | - |
| | Network | [**256FC, 256FC**] [64FC, 64FC] | - | - | - |
| | Activation Function | **ReLU**, Tanh | - | - | - |
| | Discount Factor ($\gamma$) | **0.99**, 0.97, 0.95 | - | - | - |
| | Learning Rate | **3e-4**, 1e-4, 3e-5, 1e-5 | - | - | - |
| | Tau ($\tau$) | 0.05, 0.01, 0.005 | - | - | - |
| | Entropy | **0.2**, 0.5, 1 | - | - | - |
| | Freeze Threshold | 1, 0.5, **0.1**, 0.01, 0.001 | - | - | - |

# K  Algorithm

---

**Algorithm 1** RILe Training Process

---

1: Initialize student policy $\pi_S$ and trainer policy $\pi_T$ with random weights, and the discriminator $D$ with random weights.
2: Initialize an empty replay buffer $B$
3: **for** each iteration **do**
4:     Sample trajectory $\tau_S$ using current student policy $\pi_S$
5:     Store $\tau_S$ in replay buffer $B$
6:     **for** each transition $(s, a)$ in $\tau_S$ **do**
7:         Calculate student reward $R^S$ using trainer policy:

$$R^S \leftarrow \pi_T \tag{29}$$

8:         Update $\pi_S$ using policy gradient with reward $R^S$
9:     **end for**
10:    Sample a batch of transitions from $B$
11:    Train discriminator $D$ to classify student and expert transitions

$$\max_D E_{\pi_S}[\log(D(s,a))] + E_{\pi_E}[\log(1 - D(s,a))] \tag{30}$$

12:    **for** each transition $(s, a)$ in $\tau_S$ **do**
13:        Calculate trainer reward $R^T$ using discriminator:

$$R^T \leftarrow e^{-|v(D(s,a))-a^T|} \tag{31}$$

14:        Update $\pi_T$ using policy gradient with reward $R^T$
15:    **end for**
16: **end for**

---

---

**Algorithm 2** RILe Training Process with Off-policy RL

---

1: Initialize student policy $\pi_S$, trainer policy $\pi_T$, and the discriminator $D$ with random weights.
2: Initialize an empty replay buffers $B_D$, $B_S$, $B_T$ with different sizes
3: **for** each iteration **do**
4:     Sample trajectory $\tau_S$ using current student policy $\pi_S$
5:     Store $\tau_S$ in replay buffers $B_D$, $B_S$, $B_T$
6:     Sample a batch of transitions, $b_S$ from $B_S$
7:     **for** each transition $(s, a)$ in $b_S$ **do**
8:         Calculate student reward $R^S$ using trainer policy:

$$R^S \leftarrow \pi_T \tag{32}$$

9:         Update $\pi_S$ using calculated rewards
10:     **end for**
11:     Sample a batch of transitions $b_D$ from $B_D$
12:     Train discriminator $D$ to classify student and expert transitions

$$\max_D E_{\pi_S}[\log(D(s,a))] + E_{\pi_E}[\log(1 - D(s,a))] \tag{33}$$

13:     Sample a batch of transitions, $b_T$ from $B_T$
14:     **for** each transition $(s, a)$ in $b_T$ **do**
15:         Calculate trainer reward $R^T$ using discriminator:

$$R^T \leftarrow e^{-|v(D(s,a))-a^T|} \tag{34}$$

16:         Update $\pi_T$ using calculated rewards
17:     **end for**
18: **end for**

---

