# OpenReview forum: "RILe: Reinforced Imitation Learning"
_TMLR — Rejected by TMLR_

### Review · Reviewer_drqd · 2025-05-05

**Summary Of Contributions:**

This paper proposes a new adversarial imitation learning algorithm called RILe. RILe inherits an adversarial training framework from previous works like Generative Adversarial Imitation Learning (GAIL). However, RILe modifies GAIL to include a trainer-student framework such that the adversarial reward function (discriminator from GAIL) is used as a reward function for this new trainer. The trainer now aims to learn a modified reward function that, as per the authors’ claims, (i) provides adaptive, contextual feedback to the trainer and (ii) aids exploration by motivating the student to take potentially suboptimal actions. The trainer is formulated as a POMDP and trained using model-free reinforcement learning.

The main claims of this paper are:

- The RILe procedure improves over previous AIL/IRL techniques by learning an adaptive, contextual trainer in tandem with the student policy. The authors claim that this establishes a cooperative dynamic between trainer/student and “yields a reward function that considers long-horizon effects”.
- This dynamically changing reward function is further beneficial for high-dimensional tasks where the student might need different forms of encouragement at different training stages.
- Throughout various points in the paper, the authors also claim that this procedure benefits exploration

**Audience:**

Yes

**Claims And Evidence:**

No

**Requested Changes:**

- While Figure 1 is conceptually clear to me, I find the arrows between individual sub-components quite confusing. For example, in Figure 1a, what do the arrows between the agent, env, and reward mean? It might be beneficial to label arrows like in Figure 2.
- Figure 2: From my understanding, the discriminator takes $(s^S, a^S)$ as its input. It seems more intuitive to me to pass this input to the discriminator directly from the student. Instead, here, you say that the trainer agent forwards $(s^S, a^S)$ to the discriminator. Just looking at the figure, this seems to imply that the trainer somehow modifies $(s^S, a^S)$ before passing it on to the discriminator. For better clarity, I would suggest to simply point an arrow from the student to the discriminator (and leave out the line about forwarding $(s^S, a^S)$ to the discriminator).
- I strongly urge the authors to clarify the questions in the weaknesses section. Wherever necessary, additional theoretical reasoning or experimental analyses would be highly appreciated. Please also feel free to correct me if I misinterpreted parts of the paper.

**Minor Suggestions**

> Pg1: To improve data efficiency, Adversarial Imitation Learning (AIL) methods, such as GAIL (Ho & Ermon, 2016), introduce a discriminator …
  - It seems incorrect to say that the discriminator in GAIL improves data efficiency over BC. From my understanding, the main intention is to act as a reward function for policy learning via RL. Minor rephrasing would solve this.

> Pg2: where IRL refines its reward function only after training a policy to convergence.
  - It might be nice to clarify that the policy is trained to convergence on the current reward function. The current phrasing gives a different idea.

> Pg3: We assume that we have access to m expert trajectories, all of which have n time-steps, …
  - Is it necessary for all expert trajectories to be of equal length? Why?

> Figure 1c
  - GAIL + AIRL could be interpreted as some combination of those techniques. Instead, GAIL “&” AIRL would be less confusing.

> Pg5: The discriminator distinguishes between between expert-generated state-action pairs $(s, a) \sim \tau_E$ and non-expert ones ...
  - $\tau$ is a set of trajectories. Notationally, it seems incorrect to sample individual state-action pairs from a set of trajectories. Could you please rewrite this either as sampling from the occupancy measure or indicate in a footnote that you mean you are sampling individual state-action pairs from the expert dataset.

> Pg5: represents the learned reward function that is decoupled from the environment dynamics ….
  - Please clarify what is $V_{\phi}(s)$

> Pg6: Equation (5):
  - Is there a reason why you state this as min -E[...] instead of max E[...]?

> Pg7: Formally, $A^T$ is defined as a mapping from …
  - Isn't $A_T$ just $\mathbb{R}$? And $\pi$ is the mapping from $S_T \rightarrow \mathbb{R}$?

**Strengths And Weaknesses:**

**Strengths**

- The paper is written very intuitively and clearly establishes the current state-of-the-art and the author’s intended contributions. The authors clearly position their contributions and also provide a thorough explanation of foundational concepts to this field (MDPs, RL, IRL, AIRL).
- The research problem is worth solving and the community would benefit from a technique that overcomes the drawbacks of techniques like GAIL.
- The proposed algorithm is intuitive and seems to improve notably over current baselines. Though, I do have a few reservations about it’s theoretical details and some experimental analyses.

**Weaknesses & Questions (listed in the order that they first appear in the paper)**

- The authors make two broad claims without much further elaboration or experimental support. First the authors claim that RILe’s learnt reward function considers “long horizon effects” (pg2, pg5). Could you please elaborate on what exactly you mean by long horizon effects? Does the trainer somehow implicitly learn to provide reward function strategies that go beyond the intended student learning time? Could you back this experimentally? If not, then I would recommend removing this claim.
- Second, the authors claim that RILe’s reward shaping strategy encourages exploration (pg9). This is briefly mentioned as an observation of the patterns in Figure 3. However, it is not further elaborated or rigorously analysed via experiments.
    > Pg9: RILe’s reward function dynamically adapts to the student’s current policy, providing guidance that encourage suboptimal action.


    > Pg9: Figure 3 shows RILe’s trainer encouraging exploration toward the bottom-right of the maze, which is initially suboptimal but helpful in the long run…
  - While it is intuitively clear that the reward function shown in Figure 3 encourages the agent to explore a different part of the state space (in an attempt to maximise the learnt rewards), this claim is still not analysed quantitatively. For example, are all suboptimal actions considered exploratory?  Could you mathematically elaborate on how the dynamically updated reward encourages suboptimal actions

  > Pg9: As the student learns to reach the lower part of the maze …
    - Can you comment on the effect of this non-stationarity on the training characteristics of the student in RILe? Wouldn't the fluctuating reward function lead to "forgetting" of previously learnt behaviour?

- My main concern with the proposed technique is that it does not seem very principled. The idea sounds intuitive, however, the paper lacks a clear mathematical motivation for *why* it is beneficial to learn a trainer RL agent that modifies the discriminator’s predictions. One question that goes unanswered is as follows:
    - How is $\pi_T: S \times A \rightarrow \mathbb{R}$ different from just a transformation $f(D(s_T)): S \times A \rightarrow \mathbb{R}$? Given that the trainer POMDP has deterministic transitions and the optimal trainer policy deterministically returns the max reward for the current on-policy student occupancy, shouldn’t a policy that maximises $\mathbb{E}[\exp (v(D(s_T)) - a_T)]$ be equivalent to some function of $D(s_T)$? Can this instead be expressed in closed form instead of doing RL?
    - If RL over the discriminator predictions can indeed be written as a function of the discriminator, can the trainer instead be replaced by a preprocessing (representation learning) layer on the student (such that the student takes in $D(s_T)$ and $s_T$ as inputs and processes this to learn a latent representation on which to take actions)?
    - If RL[$D(s_T)$] is indeed distinct from $f(D(s_T))$ then a mathematical justification for this would greatly cement the paper’s contributions.

- Experiments
    - The paper is missing some crucial experimental details such as the RL algorithm used to train the trainer in RILe, number of training runs per experiment, and a clear indication of variance in the various experimental results (perhaps by mentioning the standard deviation of performance across multiple runs).
    - Figure3:
        - Could you clarify what the 2D axes mean? Is it just the state space? As I understand, the reward function is a function of $s_T = (s_S, a_S)$. How does the agent's action influence the reward? Could you instead visualise the reward for a fixed action across the whole state space ($R(s_S | \text{some action } a$).
        - It also seems that these visualisations  are obtained from on-policy training. This means that one of the inputs to the discriminator (i.e the agent's on policy batch) changes between different algorithms. Hence comparing reward functions between different agent trajectories seems incorrect. Instead it would have been beneficial to compare reward functions for the same agent trajectory between all techniques. One way to do this might be to compare the offline RILe algorithm's rewards.
        - Is every episode always initialised at the same starting point? How then does training completion percentage translate to the agent's progress spatially? i.e. is it not possible that at 50% training completion the agent is still at the starting position?
        - while GAIL and AIRL maintain relatively static reward landscapes throughout training …
            - From my understanding, GAIL also takes the agent's current on-policy batch of transitions as an input for the discriminator. Why then is GAIL static throughout training?
    - Figure 4:
        - If I understand this correctly, the intention of this experiment is to measure how much the learnt rewards change over time. However, a reward function that changes constantly does not necessarily indicate that the changed reward function is an improvement over the reward function in the previous training iteration. How can you ascertain that the rewards are updated positively a each step?
        - In Fig 4b, you show that RILe has the highest L1 distance change for a set of states in the expert dataset. However, (i), shouldn't this be evaluated at the set of state-action pairs? (ii) Again, are higher reward function fluctuations on the expert states a good thing? Ideally, the agent must always be highly rewarded for visiting the same state-action pairs as the expert. Hence, a change in the reward function on the expert's state is not very indicative of a beneficial improvement to the reward function.
    - Table2: Is there a reason why the covariate shift experiment leaves out other baselines?
    - 5.3 Robotic Continuous Control: Could you elaborate on DRAIL RILe? Why does standard RILe not perform on these environments? I would be nice to get the author's ideas on failure modes of RILe.
- Discussion
    - The discussion section too seems to be missing a few crucial elements. It would have been nice to see a longer and more detailed analysis of the failure modes of RILe. Additionally, a longer discussion about its theoretical benefits over state of the art methods and potential future extensions would have been nice to see.

---

> ### Author Response · Authors · 2025-06-10
> **Response to Reviewer drqd**
>
> We thank Reviewer drqd for the positive feedback on the paper's clarity, intuitive approach, and the importance of the research problem. We appreciate the detailed questions, which will help us refine the paper's theoretical and experimental discussions.
>
> **Weakness 1: "Long horizon effects" claim**
>
> The claim that RILe's trainer considers long-horizon effects stems from the fact that the trainer itself is an independent RL agent. The trainer's objective is to maximize the *discounted sum of its future rewards*, as defined by $max_{\pi_T} E[\sum \gamma^t R_t^T]$ (Eq. 10). Its action  $a^T $ (which becomes the student's reward  $r_S $) is chosen not for myopic alignment with the discriminator's immediate output  $D(s_t^T)$, but to maximize the trainer's cumulative reward over its own learning trajectory. This incentivizes the trainer to learn a *reward-giving strategy* that has positive long-term consequences for guiding the student. For instance, it might learn to give a high reward for an imperfect but promising action if its value function,  $Q_T $, predicts this will lead to a sequence of student states that ultimately receive high scores from the discriminator.
>
> We will revise the claim on Introduction to be more precise. We will modify, "...yields a reward function that considers long-horizon effects" to "...yields a reward function which is optimized for a long-horizon objective."
>
> **Weakness 2: Claim about encouraging exploration**
>
> The claim that RILe encourages exploration is primarily a qualitative observation from experiments like the maze navigation task (Figure 3). The trainer learns to reinforce certain "suboptimal" actions only if its value function,  $Q_T $, predicts that doing so will ultimately lead to higher cumulative trainer rewards (i.e., the student eventually succeeds and gets high discriminator scores). The mathematical encouragement comes from the trainer's RL objective, which values actions based on future expected returns, not just immediate discriminator feedback.
>
> The reviewer raises an excellent point about the non-stationarity of the student's learning environment and whether this causes "forgetting." While the reward function provided by the trainer is indeed non-stationary, this is a feature, not a bug, that enables curriculum learning. The risk of forgetting is mitigated because the trainer itself is a gradually learning agent; its value function  $Q_T $ provides stability, preventing drastic immediate changes to its reward policy. Our training strategies in Appendix C,especially freezing the trainer mid-way and using a smaller buffer for the trainer to focus on recent student behaviors, are designed to mitigate excessive non-stationarity and prevent catastrophic forgetting. The goal is for the reward to adapt meaningfully to the student's learning stage, rather than fluctuating erratically.
>
> We toned down the strength of the "exploration" claim to a "qualitatively observed benefit" in our discussion of Figure 3. We also added a brief discussion on the challenge of non-stationarity in Appendix I and explicitly reference our mitigation strategies in Appendix C.

---

> > ### Author Response · Authors · 2025-06-10
> > **Response to Reviewer drqd (Continued)**
> >
> > **Weakness 3: Motivation for the trainer**
> >
> > As we also discussed with other reviewers, the trainer $π_T$ is fundamentally different from any static transformation $g(D)$ for several reasons.
> >
> > The discriminator is optimized for a *myopic, single-step classification* task (is this $(s,a)$ pair from the expert?). A static transformation function $g(D)$ would naturally inherit this myopia. In contrast, the trainer π_T is a separate RL agent optimized for a sequential decision-making problem with a *long-horizon, discounted objective* (i.e., it asks the question: “which reward sequence will change the student so that my own long-term discounted return from the discriminator is maximized?”). It learns a reward-giving *policy*, not a static function.
> >
> > The primary reason why a simple transformation cannot replace the trainer agent is that the problem is non-stationary. A fixed transformation $g(D)$ would be tied to the discriminator's myopic classification objective and apply the same reward logic regardless of the student's skill level if the discriminator is the same. In contrast, $π_T$, as an RL agent with a long-horizon objective, adapts its state-dependent reward policy based on the actual trajectories produced by the student at its current stage. To be more specific, the trainer’s value function $Q_T$ integrates the history of what reward strategies have been successful at different stages of learning. This allows the trainer to provide a reward that may seem suboptimal *now* but is advantageous for future learning. A static transformation $g(D)$ inherently lacks this foresight. Furthermore, the trainer's RL objective includes an entropy regularization term $H(π_T)$, that encourages the trainer to explore different reward regimes for any given student state. This allows it to experiment with different teaching signals, which is a mechanism that is entirely absent in direct transformations.
> >
> > Replacing the trainer with a preprocessing layer would lose this independent temporal reasoning capability explained above. The student agent already takes both $D(s_T)$ and $s_T$ in GAIL, first as the reward signal, and the second as the observation to learn which actions to take. Preserving the independent exploration and temporal reasoning based on the student’s progress with additional layers presents a distinct and challenging direction that remains an open avenue for future research.
> >
> > We included Appendix B.3 to better explain how the trainer agent's learning process and objective differ from any static transformation of the discriminator. In addition, we also included a theoretical justification of the trainer agent under Appendix B.4 by comparing it with static transformations.

---

> > > ### Author Response · Authors · 2025-06-10
> > > **Response to Reviewer drqd (Continued)**
> > >
> > > **Weakness 4: Experimental Details**
> > >
> > > * **RL Algorithm and Training Runs:** For all experiments except the maze, the trainer and student were trained using Soft Actor-Critic (SAC). We mentioned this under Appendix G in the revised manuscript. We report the number of training runs and standard errors under Appendix D and E-F, respectively.
> > >
> > > * **Figure 3:** The 2D axes in the maze environment represent the state space $s_T = (s_S, a_S)$. To visualize the reward $R(s,a)$, for each state $s_S$, we calculate reward for different fixed actions. So for each $s_S$ (for each x,y coordinate in a maze), we have multiple rewards for fixed set of actions $a_S$. We include this detail under Appendix D in the revised manuscript. The figure's intent is to qualitatively show the *difference in behavior* of the reward functions and the dynamics between the student agent and the reward function. Comparing reward functions for same agent trajectories misses the co-evaluation of the trainer-student or discriminator-student interaction, therefore we plot agent trajectories along with reward functions so that this interaction is visible. Every episode is initialized at the same starting point. The "training completion percentage" refers to the fraction of total training steps completed, not the spatial progress. As stated, GAIL also takes the agent's current on-policy batch of transitions as an input for the discriminator, but the reward function stays relatively static. This is indeed the main point we want to demonstrate with this experiment. Even though the discriminator also evolves, since it learns to *classify* expert state-action pairs, it learns quickly and remains relatively static during the training. This quasi-static reward behavior results in poor student performance in complex tasks.
> > >
> > > * **Figure 4:** The reviewer correctly notes that change does not inherently mean improvement. We ascertain that the rewards are updated positively using the Correlation between Performance and Reward (CPR) metric (Fig 4c), which shows a positive correlation between RILe's reward updates and student performance improvement. Regarding the FS-RFDC metric, we evaluate it on expert state-action pairs, not only on states, to show that RILe re-evaluates even the utility of known good state-action pairs, within the context of the student's current abilities. A high reward on an expert state the student cannot reach is an ineffective teaching signal. RILe can temporarily change this reward to incentivize the student to first learn a viable path, which is a key aspect of its adaptive nature.
> > >
> > > * **Table 2 and Section 5.3 Clarifications:** In Table 2, we chose AIRL as the most direct and relevant baseline for a reward function robustness comparison, omitting others since only AIRL claims that ‘a reward function’ is being learned. In Section 5.3, DRAIL provides a more advanced, diffusion-based discriminator. DRAIL-RILe's superior performance shows that our RILe framework is *modular and complementary* to advances in discriminator architectures. When provided with a better underlying component (a stronger discriminator), RILe's performance improves significantly. This is a strength, demonstrating that the trainer-student concept can leverage future improvements in adversarial methods. A primary failure mode for standard RILe is when the discriminator learns too quickly or is not expressive enough, causing the trainer-student dynamic to become unstable. We will add a brief discussion of failure modes in the Appendix I .
> > >
> > > **Requested Changes 1&2&3:**
> > >
> > > We appreciate the specific suggestions for improving the clarity of our figures and text. We revised Figure 1 and 2 to make the data flow clearer, as suggested. We did our best to answer all the concerns in the weaknesses above, and in the revised manuscript.
> > >
> > > **Minor Suggestions:**
> > > * **Pg1:** We rephrased the sentence in the introduction in the revised manuscript.
> > > * **Pg2:** We changed the sentence following the suggestion.
> > > * **Pg3:** It's not necessary for all expert trajectories to be of equal length for the method itself, but used for notational simplicity.
> > > * **Figure 1c:** We changed “GAIL + AIRL” to “GAIL & AIRL” to avoid confusion.
> > > * **Pg5:** We rephrased the sentence in the revision.
> > > * **Pg5:** This term represents a learned shaping term, we included a brief clarification.
> > > * **Pg6:** We changed min -E[...] to max E[...] for intuitive consistency.
> > > * **Pg7:** We removed this sentence.

---

> > > > ### Comment · Reviewer_drqd · 2025-06-13
> > > > **Response to the authors' rebuttal**
> > > >
> > > > I thank the authors for carefully addressing the concerns from my review. While the revised paper is an improvement in terms of clarity and better manages the original claims/evidence, I am still a bit unsure about a few things.
> > > >
> > > > My overall impression from the rebuttal is that the authors use the following two arguments to justify most of the concerns from my review (and some similar questions raised by other reviewers).
> > > >
> > > > - The “long-horizon” effects and exploration encouragement claims are justified by reasoning that the trainer agent uses reinforcement learning. The trainer agent aims to maximise the discounted sum of its rewards, $R^T = \exp (- | v(\mathcal{D}_{\phi}(s^T)) - a^T |)$. As per the authors, this RL training makes $\pi^T$ consider the effects of $a^T$ on its its future rewards (and by extension, the future rewards of the student).
> > > > - The entropy maximisation in the trainer objective makes it so that the trainer agent “explores different reward regimes” for the same $(s^T, a^T)$ performed by the student. The authors claim that this makes RILe more likely to find the appropriate teaching signal (something that is absent when solely using a discriminator $D_{\phi}(s^T)$).
> > > >
> > > > The authors point to the proofs in Appendix B and additionally include a few other propositions. However, my original concerns about a static transform still remain. I point the authors to lemma 2 (B.1). If I understand correctly, the lemma tries to prove that “the trainer agent learns to give higher rewards to student experiences that match with the expert state-action pair distribution”. Eq. 16 shows that the optimal trainer action at time step $t$ is $2 D_{\phi}(s^T_t) - 1$. Isn’t this just a static transform over the discriminator?  Given a student $(s^T, a^T)$ that does indeed match the expert data, doesn’t the lemma show that the optimal trainer action to encourage imitation learning (i.e. to encourage distribution matching) is to return $2 D_{\phi}(s^T_t) - 1$.
> > > >
> > > > - Yes, you do entropy maximisation in RL (and lemma 2 (B.1) doesn’t account for entropy), but even then, the optimal action would be a static transform, under entropy maximisation.
> > > > - The other concern is that you use standard policy gradient results instead of entropy regularised RL (soft value functions) results in your proofs. However, the main contribution (one of the justifications from your rebuttal) is that the entropy maximisation is a significant contribution in RILe. Given your reasoning that the entropy makes a significant contribution, ablations to test this would solidify your argument.
> > > > - For instance, what are effects of modifying the GAIL reward to be $2 D_{\phi}(s^T_t) - 1$ and then adding an entropy objective to the GAIL optimisation procedure. Comparing this with RILe should have the same empirical results.
> > > >
> > > > One might argue that the trainer agent doesn’t aim to provide “higher rewards to student experiences”. You could say that the trainer just aims to maximise its own expected return and doing this implicitly provides the student with the “correct” reward. Under this premise, lemma 2 (B.1) then is invalidated. However, even under this premise, the reasoning in section B.4 seems improper.
> > > >
> > > > $\pi^\star_T = \arg \max_{\pi_T} \mathbb{E}_{s_T, a_T}[ \gamma^t \frac{1}{\exp(| v(D) - a_T |)} ]$
> > > >
> > > > The discriminator is just an estimate of the log density ratio ($D = \log \frac{\rho_E}{\rho_{\pi_S}}$). Hence,
> > > >
> > > > $\pi^\star_T = \frac{\rho_{\pi_S}}{\rho_E} \arg \max_{\pi_T} \mathbb{E}_{s_T, a_T}[ \gamma^t \exp a_T]$
> > > >
> > > > The density ratio (being independent of $\pi_T$) now just comes out of the return, meaning that the trainer is not really optimising anything useful. I request the authors to kindly answer the questions raised above. As usual, please feel free to highlight any misunderstandings in my evaluation.
> > > >
> > > > **Firgure 3:** How do you choose which $a_S$  to use to plot the rewards.
> > > >
> > > > **Other revisions:** I appreciate the other revisions. My concerns about the experiments, and the other details are sufficiently answered.
> > > >
> > > > **Misc. Corrections:**
> > > >
> > > > - In eq. 15 $V^T(s^T_{t+1}) = \mathbb{E} [Q^T(s^T_{t+1}, a^T_{t+1})]$ . You accidentally used the s,a at the wrong time step.
> > > > - Apologies for being pedantic but in lemma 2 (B.1) after eq. 15, the line "The trainer aims to maximize $Q^T (s^T_t , a^T_t )$” should technically be “The trainer aims to take an action $a^T$ that maximises the expected $Q^T (s^T_t , a^T_t )$ …”.

---

> > > > > ### Author Response · Authors · 2025-06-16
> > > > > **Response to Reviewer drqd**
> > > > >
> > > > > We sincerely thank the Reviewer drqd for their engagement and for the insightful feedback. We have incorporated further revisions into the paper based on this feedback.
> > > > >
> > > > > **Revision of Lemma 1 (Appendix B.1):** In our original Lemma 1, we assumed a myopic objective, which creates confusion. We have revised Lemma 1 (Appendix B.1) to address this clearly. The new Lemma 1 analyzes the trainer's optimal policy under two conditions.
> > > > > * Myopic Case ($\gamma = 0$): We now discuss that if the trainer is myopic, it's policy is centered around the static transformation $a^{*T}​=2D_\phi​(s^T)−1$. This clarifies that this behavior applies only to a simplified base case that does not consider future rewards.
> > > > > * Long-Horizon Case ($\gamma>0$): We then discuss that for the full RILe trainer, where ​$\gamma>0$, the trainer agent makes a strategic trade-off between maximizing the immediate reward (which pushes $a^T$ towards $2D_\phi​−1$) and influencing the student's trajectory to achieve *higher expected future rewards* from the discriminator. This dependency on future outcomes makes the optimal action context-dependent and not replicable by static functions.
> > > > >
> > > > > We have also updated all equations in the lemma to consistently include the entropy term.
> > > > >
> > > > > **Proposed Ablation and Entropy Maximization:** This is an important point that helps clarify the distinct roles of long-horizon planning and entropy in our framework. While entropy encourages the trainer to explore different reward regimes, the primary mechanism that distinguishes RILe from a static reward function is the *long-horizon value function*.
> > > > >
> > > > > The suggested ablation (GAIL + entropy objective inside discriminator  + a reward of $2D_\phi​−1$) would still be a *static reward framework*. The reward at any time $t$ would only depend on the discriminator's output at time $t$. It lacks the main component of RILe, which is the sequential decision-making agent that uses a value function to reason about the future. RILe's trainer can provide a reward that is suboptimal *now* to guide the student to a state that will yield much higher rewards *later*. However, to empirically support this point, we have run the suggested ablation and included the resulting maze reward plots in the new Appendix J.
> > > > >
> > > > > **Trainer’s Objective:** In RILe, the discriminator output $D_\phi(s,a)$ is the direct probability from a classifier, representing an estimate of $P(expert|s,a)$, which is used as the reward inside the Bellman equation during value function update. Also, the trainer’s action $a^T$ affects the student’s policy update and therefore the distribution of next‐step transitions $(s^S_{t+1},a^S_{t+1})$. Those transitions in turn shape the discriminator’s next estimate $D_\phi(s^S_{t+1},a^S_{t+1})$, which feeds back into the trainer’s future reward $R^T$. Therefore the trainer’s Bellman updates and policy gradients remain a function of both the discriminator and the trainer’s own actions
> > > > >
> > > > > **Figure 3:** We now included the details about the actions used for plotting the rewards in Appendix D.1.
> > > > >
> > > > > **Misc. Corrections:** We corrected these sentences following the suggestion.

---

### Review · Reviewer_BZzy · 2025-05-13

**Summary Of Contributions:**

This paper proposes RILe, an imitation learning framework which proposes a student-trainer-discriminator architecture where the trainer and student agents learn cooperatively, with the trainer learning a reward function, the student learning a policy to imitate expert demonstrations (state-action trajectories) via reinforcement learning on the reward function and the discriminator being a standard student-expert state-action binary classifier similar to that in GAIL. More specifically, the trainer takes the student's state and action as input, then output a reward signal to the student as its action. The action is learned via another set of reinforcement learning where its reward function is designed as fitting the discriminator's output. All components are trained iteratively. On several experiments such as mujoco, RILe outperforms baselines such as GAIL, AIRL, IQ-Learn, etc.

**Audience:**

Yes

**Broader Impact Concerns:**

The paper does not have a broader impact statement, but I think it does not have significant concern on the ethical implications of the work (though adding a broader impact statement is of course better).

**Claims And Evidence:**

No

**Requested Changes:**

Currently, I feel some claims of the paper (especially the "cooperative" and "introduction of trainer") is not well-explained yet.

1. The paper should more clearly explain why the trainer agent is necessary (as mentioned in the weakness part), and how is the proposed method fundamentally "cooperative", since the trainer learns the reward from a discriminator competitive to the student.

2. The authors should explain more clearly on the assumption in the beginning of Appendix B: what does it mean by "the discriminator loss curve is complex"?

3. Some more details of the experiments need to be added. For example, what is the policy gradient method applied for RILe? Is it A2C, TRPO or PPO? (The authors mention "actor-critic framework" in Appendix C, and also mentions "epsilon-greedy noise"; in such case, the used algorithm should be an off-policy one.)

**Strengths And Weaknesses:**

**Strengths**

1. The experiment result is reasonable, with good visualization (e.g. Fig. 3) and a good number of ablations in Sec. 5 and appendix E,F.

2. The introduction of training tricks and hyperparameters is quite detailed, which increases the reproducibility of the paper.

**Weaknesses**

1. My biggest question is about the necessity of the trainer agent with Eq. 8. By the definition of Eq. 8, when the trainer agent is learned perfectly, it should output the **opposite** label of the discriminator's tendency as it needs to **maximize** the difference between $v(D_\phi(s^T))$ and $a_T$ (such that the trainer penalizes the student when the discriminator outputs 1, which contradicts with the text and is counterintuitive). But if the trainer reward the student when it behaves similar to the discriminator as the text and Appendix B indicates (which means negating Eq. 8), then when the trainer agent is learned perfectly, it should directly output $v(D_\phi(s^T))$ (which is exactly the lemma 1 in Appendix B). But if this is the case, why adding another layer of reinforcement learning instead of directly outputting the discriminator label helps - and how is this foundamentally different from GAIL and cooperative? Shouldn't on-the-fly reward function learning increase instability, especially at the begining of the training stage where reward is very unreliable? I understand there is an ablation in Fig. 5 which shows the design of trainer agent somehow works better, but I think this would require more careful discussion on motivation; otherwise, the experiment result can also be explained as fluctuations on a single experiment.

2. In Appendix C, the authors mention a training strategy of "freezing the trainer agent midway". But the question is: why is the trainer agent converging? If the discriminator has not converged, does that mean the discriminator can also be frozen (as it does not directly interact with the student), and will gradually unable to reflect the difference between student and expert as the student is updating but the reward signal is not? On the other hand, If the discriminator is also converging, does that mean the student has converged to the same distribution as the expert? If so, the agent is already well-trained and does not need further training with frozen trainer.

3. The proposed method has so many moving parts: trainer actor, trainer critic (as the authors mention "actor-critic framework" in Appendix C), discriminator, student actor and student critic, all trained in an iterative manner. It could be potentially quite unstable.

---

> ### Author Response · Authors · 2025-06-10
> **Response to Reviewer BZzy**
>
> We thank Reviewer BZzy for acknowledging our reasonable experimental results, good visualizations, and detailed training/hyperparameter information. We will address the concerns regarding the necessity of the trainer and the interpretation of its objective function.
>
> **Weakness 1-3 & Requested Change 1: Necessity of the Trainer Agent**
>
> The reviewer raises an essential point about the trainer's objective function, catching a critical typo in our manuscript. There should be a negative sign in the exponent of the trainer's reward function, which we have corrected in the revised paper. The correct objective is to maximize $E[e^{-|v(D_ϕ(s^T)) - a^T|}]$, which incentivizes the trainer's action a^T to match the scaled discriminator score $v(D_ϕ(s^T))$, resolving the counterintuitive interpretation.
>
> As we also explained to Reviewer iQQ8, the fundamental difference between RILe and frameworks like GAIL/AIRL lies in their core objectives. In GAIL/AIRL, the reward signal is a direct byproduct of a myopic binary classification task (Eq. 3), aimed at instantaneously separating expert and student data. Therefore, the goal is myopic classification, not strategic teaching. As we prove in Appendix B, the optimal discriminator converges to $D*(s,a) = p_E(s,a) / (p_E(s,a) + p_πS(s,a))$. The resulting reward surface often provides binary-like feedback once the discriminator becomes confident.
>
> In contrast, RILe's trainer π_T is a separate reinforcement learning agent. Its goal is to maximize a long-horizon, discounted sum of future discriminator scores (Eq. 10), meaning it learns a sequential, reward-giving policy rather than a local, static function. The trainer learns to provide a seemingly suboptimal reward at the current step if its value function, $Q_T$, predicts this will guide the student toward more expert-like states in the future. This is an ability a one-shot classifier inherently lacks. Furthermore, the trainer's RL objective includes an entropy regularization term $H(π_T)$, which explicitly encourages it to explore different reward-giving actions ($a^T$) for any given student state. This allows it to experiment with different teaching signals, a dynamic exploration mechanism that is entirely absent in GAIL/AIRL.
>
> Regarding the cooperative nature, we define the cooperation as existing between the student and trainer. The student $π_S$ works to maximize rewards from the trainer $π_T$, while the trainer works to provide rewards that guide the student to be more expert-like. Their goals are aligned: for the student to successfully imitate the expert and fool the discriminator. This cooperative pair leverages the signal from the discriminator $D_ϕ$, which remains in a competitive game with the student. We briefly discussed this in Appendix B.3 in revision.
>
> While our multi-agent setup can introduce instability, we found that the training strategies detailed in Appendix C, such as freezing the trainer and using distinct buffer sizes, are crucial for achieving stable co-adaptation and are justified by RILe's superior performance.
>
> We now include more detailed motivation in the revised manuscript (Appendix B.3), and theoretical justification of the trainer agent (Appendix B.4).
>
> **Weakness 2: Freezing the trainer agent**
>
> We freeze the trainer once its critic network within the actor-critic framework stabilizes during the training process. Critic convergence suggests that the trainer's value function, given the current student policy and discriminator, has become reliable. At this point, the trainer has learned a stable reward-giving strategy.
>
> When we freeze the trainer, we also freeze the discriminator, as its sole purpose in RILe is to provide feedback for training the trainer. Empirically, we observe that the discriminator converges much faster than the student or trainer, especially in complex tasks, since the discriminator gets better and better at classifying expert pairs from any other pairs from the beginning of the training. Its convergence, however, does not mean the student is an expert. It simply means the discriminator reliably distinguishes the student from the expert. The raw, near-binary output from an early-converged discriminator is a poor teaching signal, which leads suboptimal student performance in highly complex tasks (such as Humanoid walking in Table 4, (see answer “Necessity of Trainer Agent” for details). Therefore, the nuanced, learned policy of the trainer is still required to guide the student effectively.

---

> > ### Author Response · Authors · 2025-06-10
> > **Response to Reviewer BZzy (Continued)**
> >
> > **Requested Change 2 & 3:**
> >
> > We thank the reviewer for pointing out the typo in Appendix B. The phrase "the discriminator loss curve is complex" should indeed have been "convex." We have corrected this in the revised manuscript.
> >
> > Regarding the specific algorithm, we used Soft Actor-Critic (SAC) for both the student and the trainer agents due to its sample efficiency and stability in continuous control tasks. In Maze, we employ PPO for both. The "epsilon-greedy" hyperparameter noted in Table 8 was an auxiliary exploration mechanism we experimented with for the student, as mentioned in Appendix C, to supplement SAC's primary entropy-based exploration. We have clarified this in Appendix G of the revised paper, explicitly stating the use of SAC.

---

> > > ### Comment · Reviewer_BZzy · 2025-06-18
> > >
> > > Thanks for the clarification. Here are my responses:
> > >
> > > 1. I am still not sure what does it mean by "the loss curve is convex"; more specifically, suppose the loss function is $y=f(x)$ where $f$ is convex, what is $x$ and what is $y$? A usual interpretation of "loss curve" means is that $y$ is the loss and $x$ is the number of training steps, which apparently does not make sense in this case. Is $x$ the output of discriminator (from 0 to 1) and $y$ the loss? In this case, where is this property utilized (since there is only one "convex" appearing in the paper, which is in the assumption)?
> > >
> > > 2. After reading the author's response, my understanding is that the design of the teacher agent is based on an assumption that the discriminator is not perfect (which is also supported by the fact that it is frozen midway) due to the lack of support set by data given a fixed student. I still feel that if the discriminator is always immediately perfect after each student update, then it is fine even if the reward is myopic, because the propagation via student's Q-values will eventually pick up along the trajectories (otherwise sparse reward environments are unlearnable). But both being half-baked, the teacher agent may give a better inductive bias than the discriminator alone since the latter gives similar values to numerically similar inputs, but the former is trajectory-aware. There exists a possibility that the benefit of such inductive bias can outweigh the difficulty of training a new agent jointly, with which the point this paper holds (and "cooperative" comes from "making discriminator's reward signal more student-friendly"). This being said, the teacher still seems haunted by the lack of data as it can only train on student's state samples (which is also mentioned in the weakness part of reviewer iQQ8). There is also no proof why entropy term on the trainer agent does better in encouraging exploration than entropy term on the student agent itself.

---

> > > > ### Author Response · Authors · 2025-06-21
> > > > **Response to Reviewer BZzy**
> > > >
> > > > We thank Reviewer BZzy for the insightful follow-up. We have refined our explanations in the revised paper.
> > > >
> > > > **Convexity Assumption:** We have revised the text in Appendix B to be more precise, clarifying that this refers to the cross-entropy loss is convex with respect to the discriminator’s outputs. Our intention was to ground the argument in Lemma 2 that the discriminator can converge to its optimal version for a given set of samples by learning from the binary cross-entropy loss.
> > > >
> > > > **Role of the Trainer Agent:** The student's own Q-function is indeed designed to handle long-term credit assignment. While this works well in simple settings (such as maze), in complex, high-dimensional scenarios, a sparse or noisy reward signal fails to provide enough guidance for the student's Q-function to learn effectively. As our experiments show (Table 4), the simple binary signal from a GAIL-style discriminator leads to poor performance as task complexity increases. Furthermore, as the reviewer noted, the discriminator is often imperfect (*half-baked*). The complex environments are precisely where the trainer provides a critical benefit. The trainer agent learns to distill discriminator outputs into a richer, trajectory-aware reward and creates a dense reward landscape. This shaped reward provides useful guidance about which sub-policies are likely to lead to expert-like outcomes, making the student's credit assignment problem easier to solve. In other words, the trainer provides a dense trajectory-aware, context-dependent reward to help the student.
> > > >
> > > > The performance of the trainer is indeed tightly coupled to the data coming from the student. This is why we employ and highlight specific strategies in Appendix C to enhance the *student's own exploration*, which is particularly beneficial for RILe. Regarding the entropy, the student's and trainer's entropy terms serve complementary roles. Student entropy encourages exploration of the environment's *state-action space*, which in turn provides more diverse data for the trainer. Trainer entropy, in contrast, encourages exploration in the *space of possible reward functions*, or *teaching strategies*. This allows RILe to test different ways of rewarding the same student behavior to discover what is most effective for long-term progress.

---

### Review · Reviewer_iQQ8 · 2025-05-27

**Summary Of Contributions:**

The paper proposes RILe, a novel trainer–student framework for imitation learning that combines the strengths of inverse reinforcement learning (IRL) and adversarial imitation learning (AIL). Instead of the standard adversarial setup where a discriminator is used directly as a reward, RILe uses the discriminator to guide a trainer agent that learns a dynamic, fine-grained reward function via reinforcement learning. This reward is then used to train a student agent. The key innovation lies in the cooperative dynamic between the trainer and student, enabling more nuanced and adaptive reward shaping. RILe is validated on several benchmarks including MuJoCo and LocoMujoco, and outperforms strong baselines like GAIL, AIRL, and IQ-Learn, especially in high-dimensional and noisy settings.

**Audience:**

Yes

**Claims And Evidence:**

Yes

**Requested Changes:**

As stated in the weaknesses, I would like the authors to give a clearer explanation for the following points:

1. Please provide a clearer and more theoretically grounded explanation of the motivation for introducing the trainer agent in addition to the discriminator and student policy. While the trainer–student–discriminator structure is novel and empirically effective, it remains unclear why this architecture provides significant advantages over standard adversarial imitation setups such as GAIL and AIRL, which also adapt the reward signal throughout training.

2. In particular, clarify why GAIL and AIRL cannot achieve similar context-sensitive reward shaping, given that their generator and discriminator are co-trained. What fundamentally limits the discriminator-based reward signal in these frameworks, and how does the trainer agent overcome this limitation?

3. Additionally, the notion that the trainer can “explore reward strategies” could benefit from further elaboration. Since the trainer's reward is still derived from the discriminator, it's not immediately evident how it meaningfully expands the space of reward functions beyond what AIRL already models.

**Strengths And Weaknesses:**

**Strengths:**

1. Novelty and Conceptual Clarity:
The core idea of separating reward learning from the discriminator and introducing a trainer that co-evolves with the student is conceptually elegant and addresses key limitations of AIL and IRL. The analogy to real-world teaching (e.g., parent-child interaction) strengthens the motivation.

2. Strong Empirical Validation:
The paper performs thorough evaluations, including five well-designed ablation studies and experiments on both standard and high-dimensional continuous control benchmarks. RILe consistently outperforms baselines across diverse settings.

3. Robustness Analysis:
The authors carefully assess RILe’s robustness to noise and covariate shift, demonstrating significant resilience compared to other methods. This adds practical value.

4. Reward Adaptability Metrics:
The introduction of RFDC, FS-RFDC, and CPR provides insightful quantitative analyses of how the learned reward function evolves and aligns with performance.

5. Clear Writing and Structure:
The paper is well-written, with a logical flow from motivation to methodology and experiments. Figures (e.g., reward heatmaps) effectively convey the dynamic nature of RILe’s learned reward function.

**Weaknesses:**

While the proposed RILe framework is empirically compelling, its motivation for introducing a separate trainer agent between the discriminator and the student policy lacks sufficient theoretical or conceptual justification. The paper argues that RILe enables more context-sensitive reward shaping, but it remains unclear why such dynamic adaptation cannot be achieved by GAIL or AIRL, which also co-train the generator (student) and discriminator jointly throughout the learning process.

Furthermore, although Figures 3 and 4 empirically demonstrate that RILe learns a more dynamic and fine-grained reward function than baselines, the causal mechanism for this improvement is not well articulated. Specifically, it is not obvious why learning a reward function via a trainer agent—rather than directly from the discriminator—leads to better granularity or adaptivity in high-dimensional settings. The paper attributes this to the trainer's ability to "explore reward strategies," yet this notion is vague: since the trainer’s learning signal is still derived from the discriminator's assessment of expert-likeness, its exploration remains implicitly bounded by the support of the demonstration data, making it questionable whether true exploration is occurring.

In short, while the empirical performance of RILe is strong, the conceptual clarity around the necessity and unique benefits of the trainer agent could be significantly improved. These limitations, however, do not critically undermine the contributions of the paper. Given its strong empirical results and the novelty of the cooperative reward-learning setup, I am still leaning toward accepting the paper.

---

> ### Author Response · Authors · 2025-06-10
> **Response to Reviewer iQQ8**
>
> We thank Reviewer iQQ8 for their thorough and positive assessment of our work. We are grateful for the recognition of RILe's novelty, strong empirical validation, robustness, and clarity. The weaknesses identified are crucial, and we agree that a more detailed conceptual justification for the trainer agent will significantly strengthen the paper. We address these points in the revised manuscript.
>
> **Weakness 1 & Requested Changes 1 & 2: Justification for the Trainer Agent**
>
> The fundamental limitation of GAIL/AIRL lies in the nature of their objective functions, and RILe's trainer is designed as an RL agent to overcome this.
>
> In GAIL/AIRL, the discriminator's reward signal is a direct byproduct of a binary classification objective (Equation 3). The goal is to find a function that best separates expert data from student data at a given instant. While this function co-evolves, its goal is always *myopic classification*, not strategic teaching. As we prove in Appendix B (and as established by Goodfellow et al., 2014), the optimal discriminator converges to $D^* (s,a) = p_E(s,a) / (p_E(s,a) + p_{π_S}(s,a))$. A reward derived directly from this via generally monotonic functions (e.g., $-log(D^*)$) is a quasistatic surface that tends to saturate once the discriminator becomes confident, providing coarse, binary-like feedback.
>
> In RILe, the trainer $π_T$ is a fully-separate reinforcement learning agent whose objective is to maximize a *long-horizon, discounted sum of future discriminator scores* (Eqn. 10). Its reward $R^T$ depends on how well its action $a^T$ (the student's reward) aligns with the discriminator's score $D(s^T)$ *over time*. This establishes a crucial difference: GAIL/AIRL's reward is a quasistatic function of a one-shot classification score, whereas RILe's reward is the action of a sequential decision-maker. In other words, the trainer learns a reward-giving *policy*, not just a static function. The trainer learns to provide a seemingly suboptimal reward at step $t$ if its own value function $Q_T$ predicts this will lead to higher discriminator scores in the long run, by changing the student. This is an ability a one-shot classifier inherently lacks.
>
> Our qualitative experiment in Section 5.1 illustrates how the trainer can learn a curriculum-style reward landscape, for instance, rewarding initial exploratory actions far from the goal and later shifting rewards to fine-tune behavior near the goal, a dynamic behavior that GAIL/AIRL fails to achieve.
>
> We included more detailed motivation in the revised manuscript (Appendix B.3), and theoretical justification of the trainer agent (Appendix B.4).
>
> **Weakness 2 & Requested Change 3: Exploring Reward Strategies**
>
> When we say the trainer can explore reward strategies, we mean that because $π_T$ is an RL agent, it explores different mappings from student state-action pairs $(s^S, a^S)$ to scalar rewards $r^S = a^T$.
>
> The trainer's reward $R^T$ (Eq. 7) penalizes deviation between its action $a^T$ and the scaled discriminator signal $v(D(s^T))$. However, the trainer's RL objective includes an entropy regularization term, $H(π_T)$. This term explicitly incentivizes the trainer to explore different actions $(a^T)$ for the same student state $s^T = (s^S, a^S)$. This means the trainer doesn't just deterministically output a function of the discriminator score over the training. Instead the trainer experiments by giving slightly higher or lower rewards than the discriminator currently suggests and observe the long-term impact on the student's trajectory distribution. This is a concrete mechanism for "exploring reward strategies" that is completely absent in GAIL/AIRL, where the reward is a deterministic transformation of the discriminator's output
>
> The Fixed-State Reward Function Distribution Change (FS-RFDC) metric (Figure 4b) shows how the reward values assigned to a fixed set of expert states change over the course of training. For both GAIL and AIRL, this value is low because as the discriminator stabilizes, the reward for a given state also stabilizes. For RILe, the FS-RFDC value is significantly higher, demonstrating that the trainer is actively changing the rewards for the exact same states as the student's policy evolves. This shows how the learned reward policy adapts the reward landscape differently when compared to AIRL.
>
> We revised the method section to include the entropy regularization term. Also, explained this exploration strategy briefly under Appendix B.3.

---

> > ### Comment · Reviewer_iQQ8 · 2025-06-15
> > **Response to the authors**
> >
> > I appreciate the authors' detailed and thoughtful response. I now better understand the intended role of the trainer agent as an RL-based reward function that can adapt over time and shape student behavior with long-horizon considerations. The explanation clarifying the difference between a classifier-based reward (as in GAIL/AIRL) and a policy-based reward (as in RILe) is helpful.
> >
> > That said, I still have one conceptual question. In standard RL, the reward function is typically defined per timestep and is not responsible for encoding long-term return—this is instead handled by the value function or Q-function, which integrates future rewards. Since the student in RILe is still optimizing for the expected sum of future rewards, even a simple per-step reward can, in theory, induce long-term behavior.
> >
> > Therefore, if the trainer agent in RILe is itself learning to maximize long-term return through shaping the reward signal, its behavior starts to resemble that of a value function. It raises the interesting possibility that this trainer-based reward might be closer in spirit to a learned value estimator rather than a static reward function, potentially offering new insights into how RL structures can be augmented.
> >
> > Despite this open conceptual point, I acknowledge that the empirical results clearly demonstrate the effectiveness of the RILe framework over strong baselines like GAIL and AIRL.

---

> > > ### Author Response · Authors · 2025-06-17
> > > **Response to Reviewer iQQ8**
> > >
> > > We sincerely thank Reviewer iQQ8 for their positive feedback and this insightful conceptual point.
> > >
> > > As the reviewer stated, the trainer agent indeed learns a long-horizon value function internally and uses that value function to *shape* the reward it gives to the student. In that sense, it behaves more like a learned value estimator than a local, immediate reward. However, the trainer’s *output* remains *a per-step reward*, which the student consumes exactly as in standard RL. Another view is that RILe implements a simple form of meta-RL, where the inner learner (the student) runs off-the-shelf RL on the provided reward stream, and the outer learner (the trainer) itself runs RL to discover the optimal reward-shaping policy.
> > >
> > > All in all, RILe’s two-level structure allows us to embed the trainer's multi-step planning into a simple, per-step reward signal that the student easily consumes. This conceptual point highlights a promising direction for future research into explicitly designing and using trainer agents as dense, adaptive reward signals for complex imitation and meta-learning tasks.
> > >
> > > We appreciate the reviewer's engagement with our work and for providing this new lens through which to view our results. We now extended the discussion in the revised version.

---

### Decision · Action_Editor_tSjj · 2025-07-07

**Recommendation:** Reject

**Audience:**

Yes

**Audience Explanation:**

The idea of solving a reinforcement learning problem to find a reward function that enables the agent to achieve high (discriminator) rewards could potentially be inspiring. However, it actually seems that this overall idea is more targeted to the problem of reward shaping for reinforcement learning from sparse rewards.

**Claims And Evidence:**

No

**Claims Explanation:**

Reviewers have raised multiple concerns regarding the claims, which in my opinion have not been adequately addressed:

- The submission claims that the method matches the state-action distribution of the expert, but does not state this claim accurately (e.g. does it minimize a divergence similar to adversarial methods?), nor does it provide convincing proof. Lemma 1 considers two settings, the myopic case ($\gamma = 0$) and the long-horizon case ($\gamma \le 0$). The myopic case does not consider the entropy objective, and furthermore only shows that the reward matches a static transform of the reward, which may not be sufficient to show state-action distribution matching. The long-horizon case, only consists of a hand-wavy argument that the trainer optimizes a long-horizon objective, but does not provide a proper proof.

- The submission states that it tackles exploration by providing a more shaped reward function. However, this shaped reward function is obtained by solving a reinforcement learning problem that is framed with respect to the original discriminator rewards; hence, the trainer policy need to solve a long horizon problem, similar to the one that the policy in adversarial IL methods need to solve.

- The paper states that it optimizes a cooperative objective, however, it does not accurately state this claim. The objective of the trainer policy is different from the objective of the student policy and hence I would argue that it is not cooperative in general.

- Proposition 1 aims to motivate the trainer policy by arguing that it could produce rewards that are not a static transform of the discriminator. However, the claim is not very strong, since the mere possibility that such trainer policies exist, does not imply that the optimal trainer policies are no static transforms. Indeed, when ignoring the effect of entropy regularization, it is clear that a trainer policy that exactly follows the immediate discriminator reward also performs optimal in the long-horizon setting.

- As the paper does not provide strong claims, we need to rely on empirical evidence, which is also not convincing. The main evaluation only use a single training seed, e.g. using multiple seeds to perform several rollouts with the same policy. Furthermore, D.3 mentions that the same fixed set of seeds are used for training and validation which results in an unnecessary risk of overfitting to these seeds. The 1-sigma confidence intervals in Appendix F are often overlapping. The proposed method also employs 10 additional hyperparameters according to Table 8. In particular the "freeze thresholds" which determines when the trainer/discriminator are no longer optimized to allow for stable optimization is quite concerning.